# The interprovincial green water flow in China and its tele-connected effects on socio-economy

Shan Sang[1,2], Yan Li[1,2], Chengcheng Hou[1,2], Shuangshuang Zi[1,2], Huiqing Lin[1,2]

[1]State Key Laboratory of Earth Surface Processes and Resources Ecology, Beijing Normal University, Beijing, China

[2]Institute of Land Surface System and Sustainable Development, Faculty of Geographical Science, Beijing Normal University, Beijing, China

**Corresponding Author:**

Yan Li, Ph.D.

Institute of Land Surface System and Sustainable Development, Faculty of Geographical Science,

Beijing Normal University, Beijing, 100875, China

Email: yanli.geo@gmail.com

**Abstract:** Green water (terrestrial evapotranspiration) flows from source regions, precipitates downwind via moisture recycling, recharges water resources, and sustains the socio-economy in sink regions. However, unlike blue water, there has been limited assessment of green water flows and their tele-connected effects on socio-economy. This study used a climatology mean moisture trajectory dataset produced by the Utrack model for 2008-2017 to quantify interprovincial green water flows in China and their socio-economic contributions. Results reveal an interconnected flow network where green water of each province reciprocally exchanges with each other. Despite self-recycling (ranging from 0.6% to 35%), green water mainly forms precipitation in neighboring provinces, with average interprovincial flow directions from west to east and south to north. About 56% of total green water exported from 31 mainland source provinces remains at home, contributing to 43% of precipitation in China. The green water from source provinces embodies substantial socio-economic values for downwind provinces, accounting for about 40% of water resources, 45% of GDP, 46%

of population, and 50% of food production of China. Green water from western
provinces is the largest contributor to water resources, while green water from
southwestern and central provinces embodies the highest GDP, population, and food
production. Overall, the embodied socio-economic values of green water flow increase
from source to sink provinces, suggesting that green water from less developed
provinces effectively supports the higher socio-economic status of developed provinces.
The assessment emphasizes the substantial tele-connected socio-economic values of
green water flows and the need to incorporate them toward a more comprehensive and
effective water resources management.

### 39  1 Introduction

Terrestrial moisture recycling is a crucial process of the water cycle, whereby
water evaporates from land into the atmosphere, travels with prevailing winds, and
eventually falls back to the land as precipitation (van der Ent et al., 2010; Keys and
Wang-Erlandsson, 2018; Zemp et al., 2014). Terrestrial evapotranspiration (i.e., green
water) (Falkenmark and Rockström, 2006), which includes evaporation and
transpiration from land and vegetation, contributes to over half of the global
precipitation on land (van der Ent et al., 2010; Theeuwen et al., 2023; Tuinenburg et al.,
2020). Green water flows from upwind source regions to generate precipitation and
supply water resources for the social development of downwind sink regions through
moisture recycling (Schyns et al., 2019; Wang-Erlandsson et al., 2022). Analogous to
the upstream and downstream connection via blue water (referring to surface water and
groundwater flow within a watershed (Gleeson et al., 2020), the upwind source and
downwind sink regions are connected via green water flow within the evaporationshed
(i.e., downwind regions receiving precipitation from a specific location's evaporation)
(Ent and Savenije, 2013). Changes in both blue and green water flow directly impact
water resources availability, thereby influencing regional water security and human
societies (Keys et al., 2019).
The blue and green water flows provide a mechanism through which
upstream/upwind changes in ecohydrological and societal processes may affect the
downwind/downstream supply of water resources and, thus, ecological and societal
systems therein. Due to upstream water withdrawal and dams, global total blue water
flow into oceans and internal sinks decreased by 3.5% in 2002 compared to 1961–1990
(Döll et al., 2009). The decline in water availability exacerbated water stress in

downstream of transboundary river basins (Munia et al., 2016). Moreover, upstream vegetation restoration, soil and water conservation practices reduced water yield downstream, as already happened in the Yellow River (Wang et al., 2017; Zhou et al., 2015b). Numerous studies have investigated the causal connection of blue water flow from upstream and downstream regions, yet research into the connection of green water flow from upwind and downwind regions and their impacts remains inadequate.

Unlike blue water flow primarily shaped by terrain with specific routes and regulated by human activities (e.g., reservoir, transfer), green water flow is transported by atmospheric air movement in a pervasive manner from evapotranspiration to precipitation in downwind sink regions (Schyns et al., 2019). This establishes a spatial linkage between source and sink regions for green water flow through the moisture recycling process, similar to blue water flow through the surface hydrological process. Therefore, evapotranspiration changes associated with land cover changes in source regions are likely to impact not only downstream rivers via blue water flow but also downwind precipitation via green water flow (Keys et al., 2012), with further implications on socio-economic development (Wang-Erlandsson et al., 2018). For example, vegetation greening reduced blue water but increased downwind water availability globally through green water (Cui et al., 2022). Reduction in green water in Amazon decreased downwind precipitation in the United States (Lawrence and Vandecar, 2015), and reduction in green water source regions could decrease potential crop yields in key global food-producing regions (Bagley et al., 2012).

Source regions supply water resources to support sink regions' socio-economic development through both blue and green water flows. Existing research has extensively assessed the socio-economic values of blue water, e.g., the population dependency on runoff (Green et al., 2015; Viviroli et al., 2020), while seldom considering the tele-connected effects of green water on socio-economy. In fact, green water is also closely tied to human society because green water traveling from source regions precipitates, recharges water resources, and ultimately sustains socio-economic activities, livelihoods, and ecosystems in sink regions (Aragão, 2012; Keys and Wang-Erlandsson, 2018; O'Connor et al., 2021). These contributions should be quantified and recognized as the value of green water to socio-economy, which expands the scope of water management and water security maintenance (Keys et al., 2017; Rockström et al., 2023). Emerging moisture tracking technologies offer feasible ways to quantify green water flow across regions at large scale (Keys et al., 2019; Li et al., 2023; Theeuwen et

al., 2023) and pave the way for assessing the socio-economic values of green water.
The general spatial and seasonal patterns of moisture flows in China are
determined by regional atmospheric circulation systems, including prevailing westerly
winds (from the west toward the east) in most of China between 30° and 60° (Bridges
et al., 2023), the East Asian monsoon in eastern China, and India monsoon in
southwestern China. In summer, the East Asian and Indian monsoons supply moisture
for precipitation in eastern and southwestern China (Tian and Fan, 2013). In winter, the
East Asian monsoon drives northwesterly moisture transport across much of China and
generates precipitation (Wu and Wang, 2002). Recent studies analyzed the large-spatial
pattern of moisture recycling in China at the grid (Zhang et al., 2023a), river basin
(Wang et al., 2023b), and ecological regions scales (Xie et al., 2024), or for specific
regions (Pranindita et al., 2022; Zhang et al., 2024). However, green water flows from
different regions are interlinked and become sources and sinks of each other. Such green
water transfer at a sub-national scale effectively forms an interconnected green water
flow network. It highlights the mutual dependency of green water and its socio-
economic contributions, especially for large countries like China. Few studies focus on
green water flows at the administrative district scale, which is important for water
management. Furthermore, the substantial regional disparities in socio-economic
development add complexity to understanding the socio-economic contributions of
green water among Chinese provinces. The western provinces with a weak economic
status and sparse populations are abundant in water resources (Ya-Feng et al., 2020). In
contrast, the economically developed and densely populated eastern provinces suffer
from water scarcity (Varis and Vakkilainen, 2001). Therefore, quantifying
interprovincial green water flows and evaluating the embedded socio-economic values
offer new perspectives for optimizing water resource utilization and mitigating the
imbalance in regional socio-economic development.
In this study, we used a high-quality moisture trajectory dataset from the UTrack
model to quantify and visualize the interprovincial network of green water flows within
China. Next, we combined socio-economic statistical data to evaluate socio-economic
values embodied in green water flow for economic production, population and food
production. Our study aims to reveal the transboundary green water flows within China
and their tele-connected effects on the socio-economy. This study incorporates green
water flow into water resources, extending water resources management beyond blue
water toward a more complete understanding of the water cycle and its socio-economic
implications, which is beneficial to assess and optimize regional water security.

## 2 Data and Methods

### 2.1 Data

This study used the moisture trajectory dataset generated by the Lagrangian
moisture tracking model "UTrack-atmospheric-moisture" driven by ERA5 reanalysis
data. The model is the state-of-the-art moisture tracking model, producing more
detailed evaporation footprints due to the high spatial resolution and reduced
unnecessary complexity (Tuinenburg and Staal, 2020). The dataset provides monthly
mean moisture flows at the global scale with a spatial resolution of 0.5° for 2008-2017,
expressed as the fractions of evaporation from a source grid allocated to precipitation
at a sink grid (Tuinenburg et al., 2020). It has been widely used in moisture recycling
research with various spatial scales, such as precipitation source of the grid (Staal et al.,
2023; Wei et al., 2024; Zhang et al., 2023a) and basin scale (Wang et al., 2023b), and
moisture transport between nations (Rockström et al., 2023). The moisture trajectory
dataset was used in conjunction with the multi-year monthly mean ET of 2008–2017
from the ERA5 reanalysis dataset to estimate precipitation in a sink grid originating
from a source grid.
The socio-economic statistical data in 2008-2017 from the China Statistical
Yearbook were used to estimate the socio-economic values of green water in terms of
water resources volume, gross domestic product (GDP), population, and food
production for 31 provinces in mainland China, without Hong Kong, Macau, and
Taiwan due to the data limitation. GDP was adjusted to price in the year 2020 to
eliminate the effects of inflation.

### 2.2 Quantify green water flows in China

We quantified interprovincial moisture flows and their precipitation contribution
following the workflow described in Fig. A1. At each sink grid, the evapotranspiration
(ET) to precipitation (ET-to-P) fractions from the moisture trajectory datasets were
multiplied by ERA5 ET to obtain monthly precipitation contribution by moisture from
its source grids. Repeating the calculation for all grids within a sink province and
summing them up yielded the precipitation in the sink province contributed by each
source grid (Fig. A1 Step 1). Next, we employed zonal statistics to sum up precipitation
in the sink province contributed by grids of each source province, and the precipitation
contribution was converted to relative values, i.e., the fraction of precipitation in sink
province $j$ originating from green water of a source province $i$ (denoted as $W_{ij}$) rather
than absolute contribution to reduce the uncertainty in the latter (Fig. A1 Step 2). The
fractions $W_{ij}$ multiplied by the observed precipitation of the sink province restore the
absolute precipitation contribution. This practice ensures that provincial precipitation
is fully decomposed into different sources, reducing the estimation bias of sink
precipitation due to unclosed water balance by ET and precipitation data (De Petrillo et
al., 2024). Finally, the interprovincial green water flows in China were derived after
estimating each province individually.
The direction of green water flows can be represented by a vector starting from a
source to sink province determined by their geometric centers and with its length
denoting flow magnitude. Since green water flows have multiple destinations, each
flow points to different sink provinces, and even outside of China. For each source
province, all of their domestic green water flow vectors can be averaged to a composite
to represent their net direction and magnitude, which are mainly determined by
atmospheric wind conditions, source location and green water volume.

## 2.3 Quantify socio-economic values embodied in green water

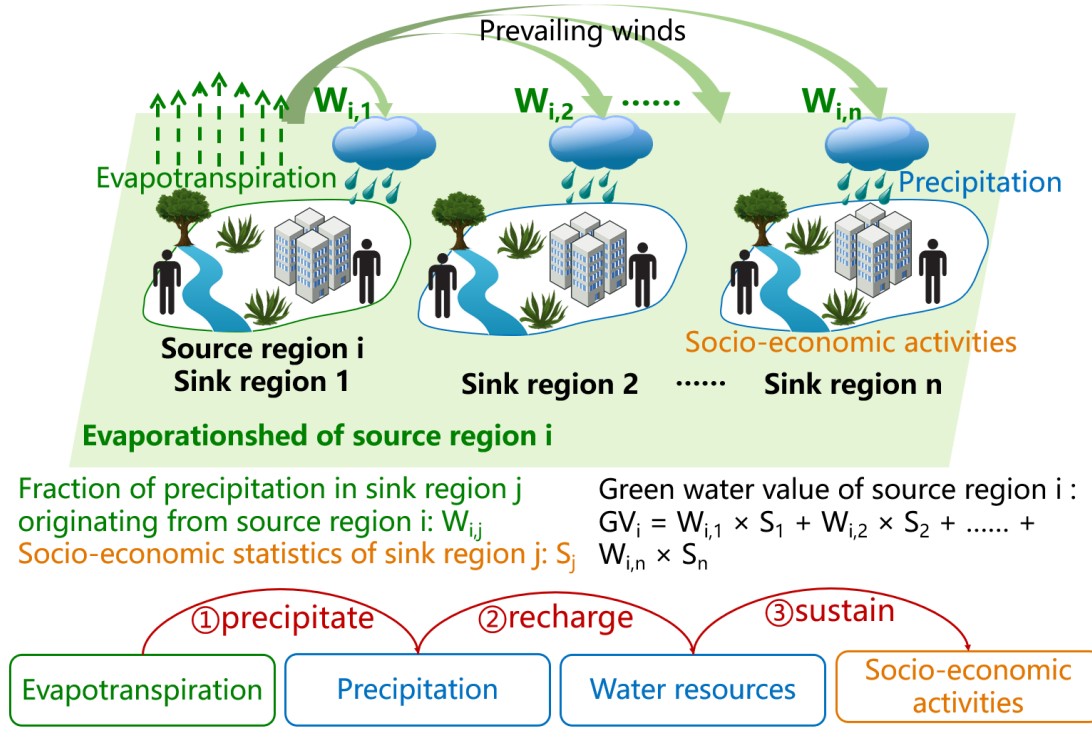

Figure 1. A conceptual diagram depicts the teleconnection of green water flows and their socio-
economic contributions in a cascade manner. Evapotranspiration (green dotted arrows) from
source region *i* flows downwind with prevailing winds (green thick arrows) and precipitates in
sink region *n*, which recharges water sources and sustains socio-economic activities in sink
regions.
Green water from upwind source provinces flows and precipitates downwind to
recharge water resources, and therefore sustains socio-economic activities in sink
provinces, as depicted in Fig. 1. Consequently, precipitation, water resources, and
socio-economic factors such as economic activities, human livelihood, and crop
production in sink provinces rely on green water exported from source provinces.
Changes in green water may affect water resource volume, and then impact economic
activities, livelihood, and crop production through water supply. We chose water
resources volume, economic output (measured by GDP), population, and food
production as the four socio-economic indictors that are tightly related to water
resources to evaluate the socio-economic contributions of green water.
If we assume all socio-economic activities in sink province $j$ are sustained by
precipitation which constitutes water resources and recharges groundwater, socio-
economic statistics of sink province $j$ can be partitioned to source provinces by their
share of precipitation contribution ($W_{ij}$). Therefore, multiplying socio-economic
statistics in sink province $j$ ($S_j$) by $W_{ij}$ yielded the socio-economic value of green water
from source province $i$. The total socio-economic value of green water of source
province $i$ ($GV_i$) can be obtained by summing its contributions to all sink provinces (Fig.
1), as equation (1):
$$GV_i = \sum_{j=1}^{n} (W_{i,j} \times S_j), \qquad (1)$$
where $S_j$ is the average socio-economic value of 2008-2017 (i.e., water resources
volume (km$^3$), GDP (in unit of CNY, 1 CNY = 0.14 USD), population (persons), and
food production (ton)) at sink province $j$, $n$ is the number of sink provinces.
Due to the different socio-economic development statuses, the same amount of
green water may produce different socio-economic values between source and sink
provinces. This means green water flow also involves changes in embodied socio-
economic value from source to sink provinces. We used water productivity in the source
province ($WP_i$) to calculate the socio-economic values of its exported green water in
the counterfactual scenario when it was all consumed in the source province without
interprovincial transfer ($GV_i'$) (Eq. 2). The results were compared with the actual green
water's socio-economic values (Eq. 1) (namely socio-economic values of exported
green water when it is consumed in sink provinces) as:
$$GV_i' = \sum_{j=1}^{n} (W_{i,j} \times WU_j \times WP_i), \qquad (2)$$
where $WU_j$ is water use in sink province $j$, and $WP_i$ is water productivity in source
province $i$. (i.e., economic output, population, and food production per unit water use).
The changes in the socio-economic value of green water flow ($\Delta GV_i$) from source

province *i* to its sink provinces can be estimated by Eq. 3.

$$\Delta GV_i = GV_i - GV_i'$$ (3)

$\sum_{i=1}^{n} \Delta GV_i$ is the net change in socio-economic values of all interprovincial green water flows in China.

## 3 Results

### 3.1 The interprovincial green water flows in China and their directions

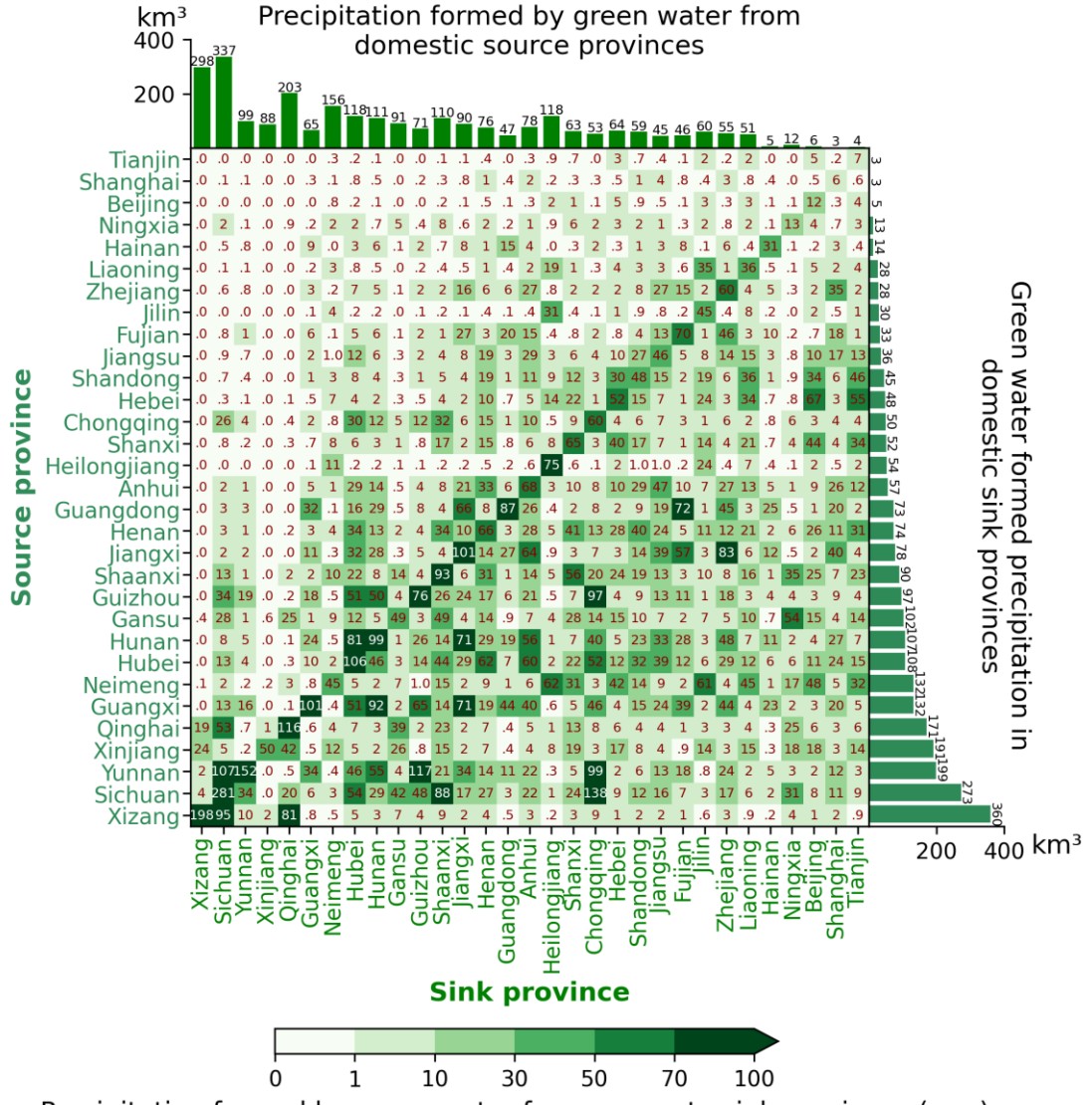

Figure 2. Interprovincial green water flows in China. The heat map denotes precipitation in sink province generated by green water from a source province (mm). The right bar shows domestic precipitation (km³) formed by green water from each source province. The top bar shows precipitation in each sink province formed by green water from domestic source provinces (km³).

Green water exported from a source province forms precipitation in different sink provinces in China, and precipitation in a sink province originates from green water in

different source provinces. Therefore, different provinces in China, acting either as
sources or sinks, are interconnected through moisture recycling and established an
interprovincial network (Fig. 2).
A large fraction of green water exported from each source province is retained
locally to generate precipitation (diagonal cells in Fig. 2). The precipitation recycling
ratio (PRR), the ratio of precipitation generated by local green water to total
precipitation, reflects how much green water of each source province contributes to its
own precipitation (Fig. A2c). Xizang has the highest PRR of 0.345, followed by
Qinghai (0.341) and Sichuan (0.297). Besides local recycling, green water
predominantly flows and generates more precipitation in neighboring provinces and
less in distant provinces. For example, green water from Sichuan forms high
precipitation in neighboring provinces such as Chongqing (138 mm), far surpassing
other distant sink provinces (< 88 mm).

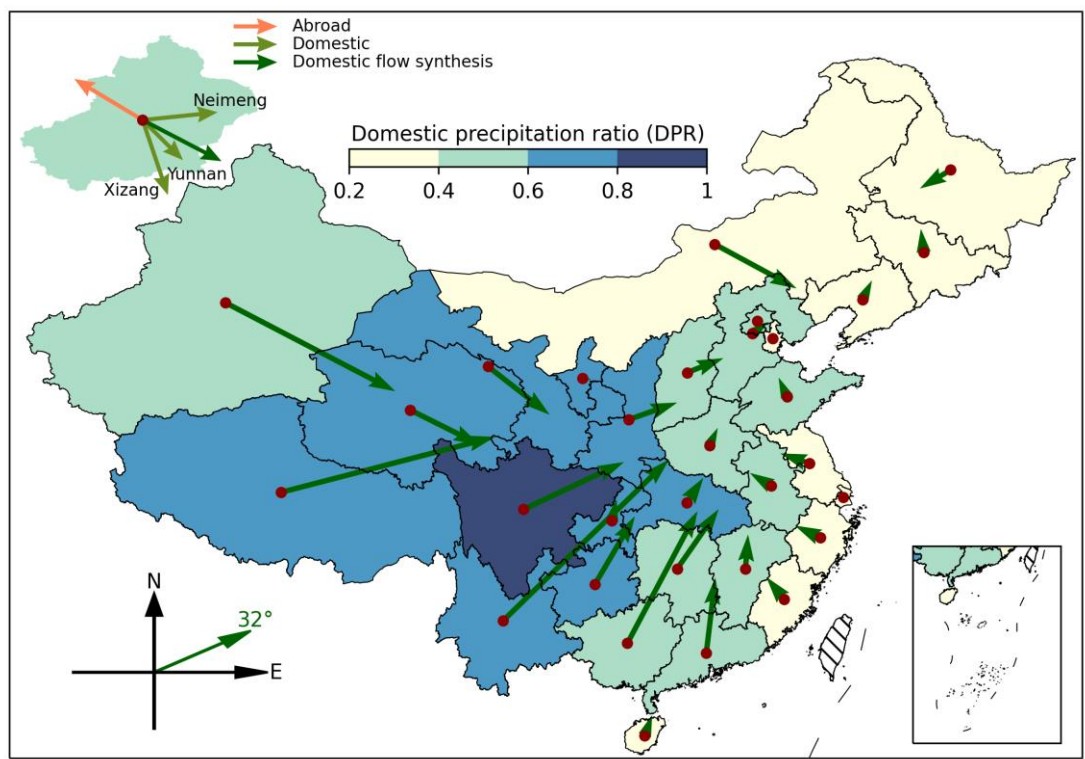

Figure 3. Direction of green water flows from each source province in China. Green arrows
indicate the average direction of domestic green water flows, denoted as a vector starting from a
source (the geometric center in red points) to sink provinces and with its length representing the
amount of precipitation formed by green water. The face colors on the map represent fractions of
green water formed precipitation within China of each source province (DPR). The upper left
corner is a schematic diagram for green water flows from Xinjiang. The lower left corner is the
254        composite flow direction of interprovincial green water of all provinces.

The direction of interprovincial green water flow can be visualized as a composite
direction averaging all domestic green water flows from each source province, which
are mainly determined by atmospheric wind conditions, source location, and green
water volume (Fig. 3). Overall, the average direction of all interprovincial green water
flows is at 32° northeastward (32° north off the east direction), suggesting green water
within China is transported to the north and east directions owing to combined effects
of monsoons and westerly.

262       Green water exported by source provinces contributes to precipitation both within

and outside China. We defined the domestic precipitation ratio (DPR) as the ratio of
green water that formed precipitation in China to each province's total green water
export to represent their relative importance to China's precipitation (Fig. A2a). Green
water from provinces in western and central China mainly flows eastward under the
influence of prevailing westerlies, which extend their evaporationsheds eastward to
cover a large territory of China and generate more precipitation within China (Fig. 3).
For instance, green water from Xizang, the largest exporter in China, produces the
largest domestic precipitation (360 $km^3$) (right bar on Fig 2) with a high DPR of 0.74,
contributing to precipitation in other 30 provinces with varying extents (0.2 to 95 mm).
Similarly, the green water from southern provinces is affected by the Indian Ocean
Monsoon (southwest monsoon), which drives green water flowing northeastward. With
a substantial volume of green water, these southern provinces contribute significantly
to domestic precipitation. In contrast, green water from eastern coastal or northwest
border provinces goes to the northwest primarily attributed to the East Asian Monsoon
(southeast monsoon) (Cai et al., 2010). As a result, most evaporationsheds laid outside
China generate less domestic precipitation but more outside the country, resulting in a
lower DPR, such as Fujian (DPR 0.31) and Heilongjiang (DPR 0.23). The northern
provinces are influenced by westerly winds and winter monsoon from Siberia (Sun et
al., 2012), causing predominantly southeastward flow of green water. However,
evaporationsheds of these provinces mainly cover the Pacific Ocean, resulting in a
relatively low DPR despite their substantial volume of exported green water. While
some inland provinces have a high DPR because their evaporationsheds overlap with
mainland China, the low green water volume (Fig. A4) limits their domestic
precipitation contribution (e.g., Gansu and Ningxia with DPR of 0.72 and 0.66,
respectively).

288       Furthermore, precipitation in sink provinces originates from both domestic and

foreign green water sources. Sichuan (337 $km^3$), Xizang (298 $km^3$), and Qinghai (203

km³) are the top 3 provinces importing the largest volume of green water from domestic sources due to the large ET from themselves and neighboring provinces (top bar of Fig 2). To quantify the relative importance of domestic sources, we defined the domestic source ratio (DSR) in each province as the sum of precipitation contribution from domestic sources divided by total precipitation (Fig. A2 (b)). DSR is related to each province's precipitationshed (i.e., upwind region contributing evaporation to a specific location's precipitation) (Keys et al., 2014) and the included domestic green water exporters. The highest DSR found in Qinghai (0.86) and Ningxia (0.82) is because their precipitationsheds include large domestic green water exporters like Xinjiang and Xizang, which supply considerable green water traveling eastward. Conversely, Hainan (0.07) and Guangdong (0.14) in coastal areas have lower DSR because their precipitationsheds are primarily located in oceans and other countries due to the influence of the summer monsoon (Cai et al., 2010).

## 3.2 Socio-economic values embodied in interprovincial green water flows

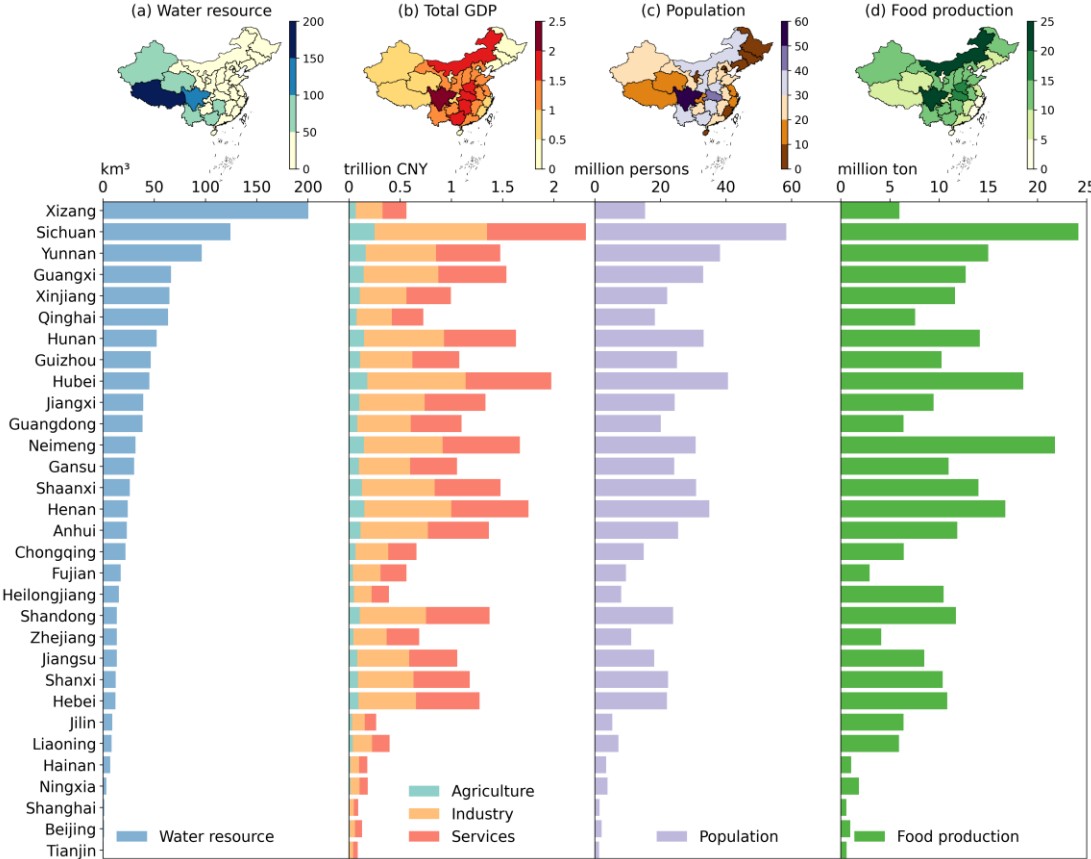

Figure 4. The embodied socio-economic values of green water flow from source provinces for water resources, GDP, population, and food production (average value of 2008-2017) of sink provinces in China.

Source provinces export green water and create precipitation to sink provinces
through moisture recycling process, recharging water resources and sustaining the
socio-economic development of downwind sink provinces (Fig. 4). The reliance of
socio-economic activities in sink provinces on green water supply from source
provinces implies that the green water and socio-economy are intertwined through the
interprovincial green water flow network, indicating a teleconnection between source
and sink provinces.
Our assessment of contribution of green water to water resources indicates that
green water from western provinces recharges the highest volume of water resources.
Xizang (200 km$^3$), Sichuan (124 km$^3$), and Yunnan (96 km$^3$) are the top 3 contributors
of water resources, whose green water export makes up 46%, 51%, and 52% of their
own total water resources, respectively (Table. A1). These regions also correspond to
the top contributors to domestic precipitation, owing to the close linkage between
precipitation and water resources. Although southern and eastern provinces are rich in
water resources due to the wet climate, most of their green water contributes to water
resources outside of China or to the ocean since they are situated downwind of
prevailing westerlies and proximate to the coast (e.g., Guangdong). In total, green water
exported from 31 provinces contributes 43% and 40% of precipitation and water
resources in China (Table. A1).
The GDP, population, and food production embodied in green water export from
source provinces are shown in Fig 5b-d, which reflects how much the socio-economy
of downwind sink provinces is supported by green water of source provinces. Overall,
the contribution of green water to selected socio-economic statistics shows similar
rankings because food production and agriculture GDP (R = 0.79), population and total
GDP (R = 0.85) are spatially correlated (Fig. A6).
Sectoral GDP embodied in green water from source provinces is highly related to
the industrial structure in sink provinces. The embodied industry and service sector
GDP values across provinces are relatively comparable, whereas embodied agricultural
GDP values are lower due to the small percentage of agricultural output to total GDP
(Fig. A3).
Green water from southwest and central provinces (e.g., Sichuan, Hubei, Henan)
embodies the most GDP, population, and food production because of the large
economic volume of these provinces and neighboring regions, as well as the high DPR.
Specifically, green water from Sichuan supports the highest GDP (2.31 trillion CNY),
population (58 million persons), and food production (24 million tons) (Table. A2)
because Sichuan has a high GDP, population, and food production (Fig. A3). Moreover,
green water from Sichuan contributes significantly to its own precipitation (30%), and
87% of its green water generates domestic precipitation. These factors together make
green water in provinces like Sichuan embody the highest socio-economic values.
Provinces that export large volumes of green water and have high DPR do not
necessarily embody more socio-economic values if sink provinces that import their
green water are less developed. Xizang is the highest green water exporter and the
largest contributor of water resources (200 km$^3$) but ranks low in embodied GDP (0.56
trillion CNY, 23$^{rd}$), population (15 million, 20$^{th}$), and food production (5.97 million tons,
23$^{rd}$) because the primary importer of its green water, such as Xizang and Qinghai, have
low rankings in GDP (31$^{st}$, 30$^{th}$), population (31$^{st}$ and 30$^{th}$), and food production (30$^{th}$
and 29$^{th}$).
Green water from highly developed provinces (e.g., southeastern China) may not
necessarily embody high socio-economic value if they have low DPR. For example,
Guangdong ranks 1$^{st}$ in GDP and population and 17$^{th}$ in food production but only has a
small fraction of green water contributing to domestic precipitation (DPR 0.4). The
limited domestic precipitation contribution results in low rankings of embodied socio-
economic values (14$^{th}$ for GDP, 17$^{th}$ for population, and 21$^{st}$ for food production) for
Guangdong.
## 3.3 Changing socio-economic values of green water flows

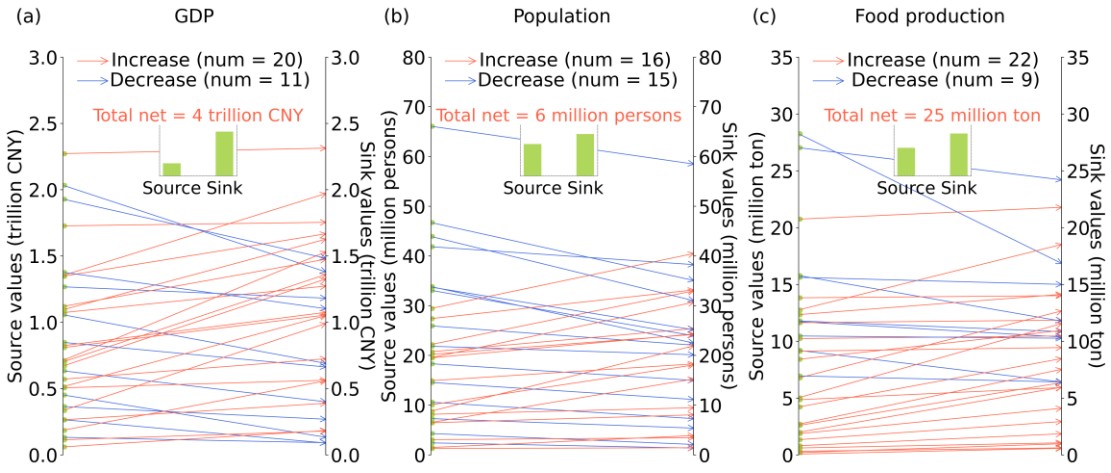

Figure 5. Changes in socio-economic values embodied in green water flow from source to sink
provinces for GDP (a), population (b), and food production (c). Thin arrows of different colors
represent the socio-economic value increase (in red) or decrease (in blue) from source to sink
provinces. Green bars represent the sum socio-economic value in China's 31 provinces.
The substantial socio-economic values embodied in interprovincial green water
flows highlight the teleconnection of green water from source provinces and the socio-
economy in sink provinces, including economy, population, and food production. Due
to different socio-economic statuses, the same amount of consumed water resources,
which are recharged by green water, would sustain different socio-economic values
between source and sink provinces. Therefore, the socio-economic values embodied in
green water flow would change when traveling from source to sink provinces. As shown
in Fig. 5, the socio-economic values embodied in green water flow increase from source
to sink provinces by 4 trillion CNY for GDP, 6 million for population, and 25 million
tons for food production, respectively. The increase in the embodied GDP, population,
and food production is observed in 20, 16, and 22 source provinces among a total of 31.
This indicates that green water tends to flow from less to more developed provinces,
sustaining more economic production, population, and food production per unit of
green water. The largest economic output value increases are in Guangxi (+0.83 trillion
CNY, 54%). Xinjiang has the most added value in population (+13 million persons,
59%) and food production (+7 million tons, 60%) because their green water flows to
more developed provinces (Fig. A5). In contrast, decreased socio-economic values of
green water flow are also observed. Shandong, Shaanxi, and Henan have the largest
depreciation in green water values for GDP (-0.66 trillion CNY, 48%), population (-13
million persons, 42%), and food production (-12 million tons, 72%) (Fig. A5) because
their green water flows to provinces with lower socio-economic values.
The changing socio-economic values of green water flow reflect the regional
disparity in socio-economic statuses between source and sink provinces. The exported
green water for more than half of the source provinces in China (> 15) has increased
socio-economic values when reaching sink provinces. This shows that green water from
less developed provinces effectively supports the higher socio-economic status of
developed provinces through the interprovincial flow network. Therefore, these
provinces are vitally important green water providers to developed areas. This
teleconnection of green water and socio-economy substantiates that changing land use
in the source provinces that affect evapotranspiration is likely to influence water
resources availability and socio-economic development in the sink provinces (Dias et
al., 2015; Weng et al., 2018). Hence, it is imperative to account for "invisible" green
water flow and its cascade effect in large-scale water resources management.
**4 Discussion**
This study quantified the interprovincial green water flows in China using the

moisture recycling framework and a moisture tracking model. The green water flow is established by transporting evaporated moisture by atmospheric winds from a source province to precipitate in a sink province. The transferred green water exchanges among multiple provinces and creates an interprovincial flow network. The location of the source province and its flow direction largely determine to what extent green water formed precipitation retains within China. In our estimation, roughly 43% of green water forms precipitation in China, similar to 44% of PRR identified by Rockström et al. (2023). The average direction of all interprovincial green water flows in China is from southwest to northeast, consistent with findings by Xie et al. (2024).

Green water flow can fill the gap in land-atmosphere feedback in the traditional water resources management framework (Keys et al., 2017). Typically, water resources management only considers blue water changes while neglecting green water flow, even though the latter may compensate for the former (Hoek van Dijke et al., 2022). Human activities such as irrigation (Su et al., 2021), afforestation (Li et al., 2018), and reservoir construction (Biemans et al., 2011; Veldkamp et al., 2017) in upstream regions may markedly change blue water accessibility in downstream regions. Meanwhile, the resulting changes of ET in upstream regions (McDermid et al., 2023; Qin, 2021; Shao et al., 2019) might offset the decline of water resources in downstream by moisture recycling. Similarly, increased vegetation coverage intercepts more rainfall, reducing runoff and consequently diminishing water resources availability (Sun et al., 2006; Zhou et al., 2015a), but the rise of ET may compensate for local and downwind water availability through increased green water flows (Wang et al., 2023a; Zhang et al., 2021). Therefore, green water is an essential path of climatic and hydrological interaction among different regions, providing a new angle for integrated regional resources management (Keys et al., 2018; te Wierik et al., 2021). A comprehensive impact assessment of regional water security and optimization would benefit from combining both blue and green water flows (Schyns et al., 2019) by which upstream/upwind regions affect regional water resource availability (Creed et al., 2019).

With the recognition of the tele-connected effects of green water flows, maintaining regional water security requires both rational utilization of local water resources and appropriate land management in the upwind source regions. However, similar to blue water, water resource management across administrative boundaries has always been challenging due to conflicting interests among different regions (Rockström et al., 2023). The diverse strategies developed to enhance regional

coordination of blue water management serve as a reference for green water management, such as the inter-basin water transfer or downstream beneficiaries paying upstream providers for clean water services (Farley and Costanza, 2010; Pissarra et al., 2021; Sheng and Webber, 2021). However, unlike blue water resources with well-established accounting and valuation methods, green water monitoring and valuation are challenging. Green water from a specific region flows to multiple regions, and the received green water can subsequently reevaporate and flow to other regions (Zemp et al., 2014). This interconnected network and cascade complicate the quantification of how much green water from a source region contributes to human activities in sink regions. More importantly, it is difficult to measure green water flow through observations as those measurements made by hydrologic stations for blue water (Hu et al., 2023; Sheng and Webber, 2021). This study utilized a dataset from a moisture tracking model to construct an interprovincial green water flow within China, which offers valuable insights for understanding the quantity of green water flow.

Due to the complex dynamics of the green water flow and limitations of the moisture tracking model, there are still major uncertainties in data and methods of this study. First, ET and precipitation datasets driving the UTrack model affect the tracked trajectories and magnitude of moisture flow. The resulting moisture trajectory is expressed as the ET-to-P fraction, and the exact amount of moisture is restored by the ET and precipitation datasets chosen by users. Different ET and precipitation datasets could lead to different precipitation contributions and PRR (Li et al., 2023). We used the ERA5 dataset to keep consistent with the original UTrack model. It is noted that the non-closure of the hydrological balance from ERA5 (De Petrillo et al., 2024) and divergence in moisture tracking models (e.g., simplifications and assumptions) also add uncertainty and impact the accuracy of the tracked green water flow (Tuinenburg and Staal, 2020; Zhang et al., 2023b). Moreover, the resulting moisture trajectory data only represent the climatologically average moisture trajectories and ET (Li et al., 2023), neglecting the interannual variability in moisture flow trajectory, e.g., those induced by the influence of extreme weather events or ENSO (Zhao and Zhou, 2021). The interannual variations in green water flow may affect DPR and DSR in some provinces. Human adaptation tends to buffer the impacts of interannual variations on the socio-economy through water resource management such as reservoirs, dams, and other infrastructure. Accounting for interannual variations in green water flows and their socio-economic contribution is worth further investigation. Secondly, the socio-

economic value assessment of green water in this study only considers green water
flows within China, excluding flows moving abroad and to the ocean that may embody
socio-economic value beyond the territory of mainland China. We mainly attribute
socio-economic values to green water and generated precipitation because precipitation
is the ultimate water source for recharging surface and groundwater of a region. Strictly
speaking, such attribution needs to be more precise because socio-economy also utilizes
streamflow from upstream areas, which deserve separate attention.
Moreover, the interactions between blue and green water increase the complexity
to evaluating green water's socio-economic contribution. For example, the blue water
extracted by irrigation increases ET in the source region, providing more moisture for
downwind regions (Yang et al., 2019). Simultaneously, most of the blue water for local
irrigation comes from the green water of upwind regions (McDermid et al., 2023). In
addition, not all water resources replenished by green water-induced precipitation are
accessible for human activities since part of them is used by the natural ecosystem
(Keys et al., 2019). Therefore, it is necessary to distinguish water sources and
consumption to account green water values more accurately. Despite the selected socio-
economic indicators closely linked to water resources, green water flows' socio-
economic contribution can manifest in other aspects such as livestock production and
irrigated agriculture. In future studies, the dynamic linkage between green water, water
resources and economic development can be assessed annually by using a long-term
moisture tracking dataset with a separation of water sources consumed by socio-
economy (surface and groundwater). Nevertheless, our assessment serves as a useful
first step to demonstrate the importance of the tele-connected green water flow in
addition to blue water. Our attempts to quantify the socio-economy embodied in green
water flow fill the gap in green water value assessment and provide a methodological
reference for green water management.

**5 Conclusion**

This study quantified the interprovincial green water flows in China and its tele-
connected effects on the socio-economy. The green water exchanges among different
regions effectively form a complex flow network and embody socio-economic values.
The interprovincial green water in China flows primarily from west to east and to a
lesser extent from south to north, influenced by the co-control of westerlies and
monsoons. Western provinces have significant contributions to precipitation and water
resources in China, while southwestern and central provinces have the most socio-

economic values regarding GDP, population, and food production. Green water flowing from less developed regions supports substantial socio-economic values in more affluent regions due to disparity in socio-economic development between source and sink regions. Given the embodied socio-economic benefits of green water, regional water resources management should consider water flow beyond blue water to integrate green water for a more comprehensive and effective management of resources and security. Our study provides a reference for understanding the "invisible" green water flow and its tele-connected benefits.

## Data and code availability

The moisture trajectory dataset is available at https://doi.pangaea.de/10.1594/PANGAEA.912710 (Tuinenburg et al., 2020). The evapotranspiration data from ERA5 reanalysis dataset is available at https://cds.climate.copernicus.eu/datasets/reanalysis-era5-single-levels-monthly-means?tab=overview (Hersbach et al., 2023). The socio-economic statistics data is available from China Statistical Yearbook (https://data.stats.gov.cn/index.htm).

The Python codes and data used in this study are available at GitHub (https://github.com/sangshan-ss/GW-China).

## Author contributions

YL and SS conceived the study and performed data analysis. SS and YL wrote the manuscript with contributions from CCH, SSZ and HQL.

## Competing interests

We declare no conflict of interest of this work.

## Financial support

This research was funded by the National Natural Science Foundation of China (42041007), the Second Tibetan Plateau Scientific Expedition and Research Program (2019QZKK0405) and the Fundamental Research Funds for the Central Universities.

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

# Appendix A

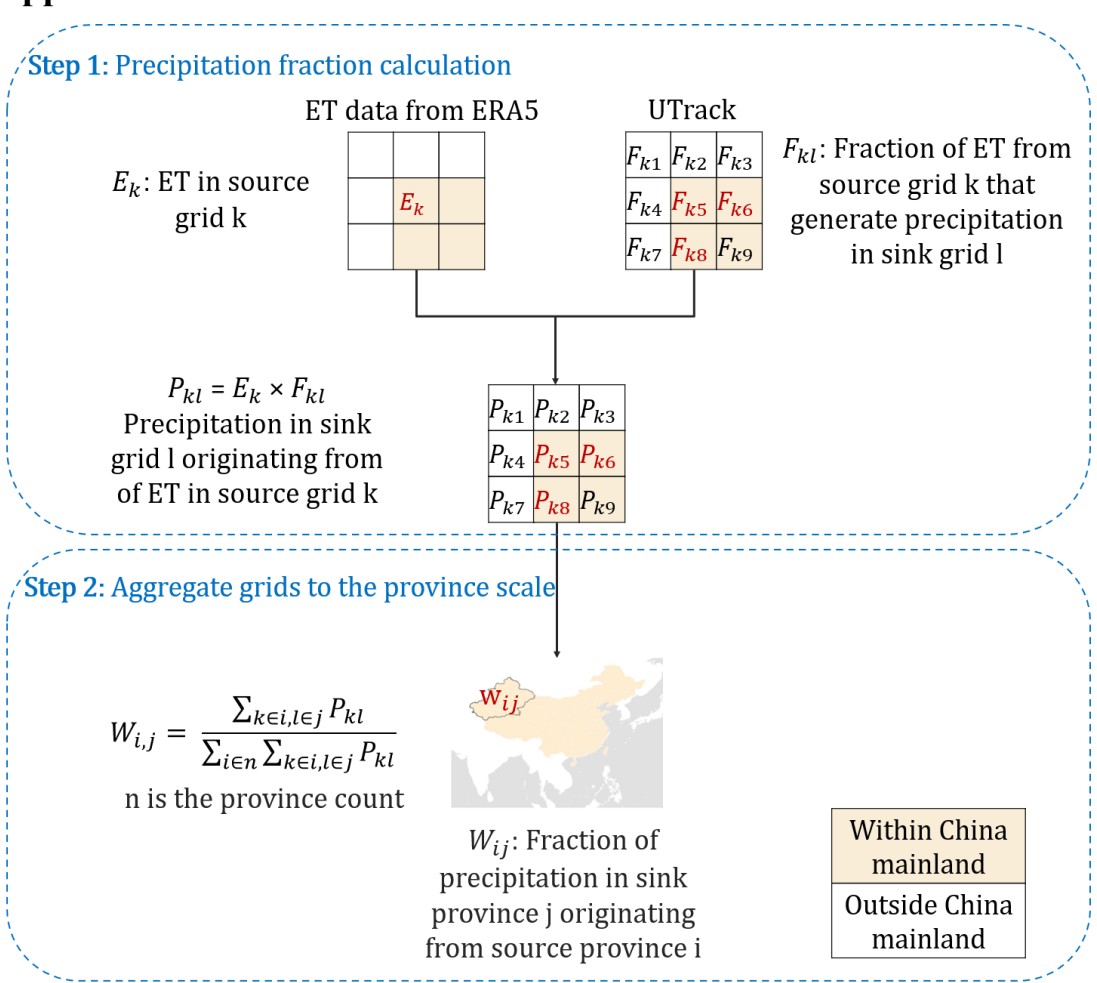


**Figure A1.** Workflow of estimating green water flow. Step 1: calculate precipitation in sink grids
originating from ET in source grids. Step 2: calculate the fraction of precipitation in sink
provinces originating from source provinces.

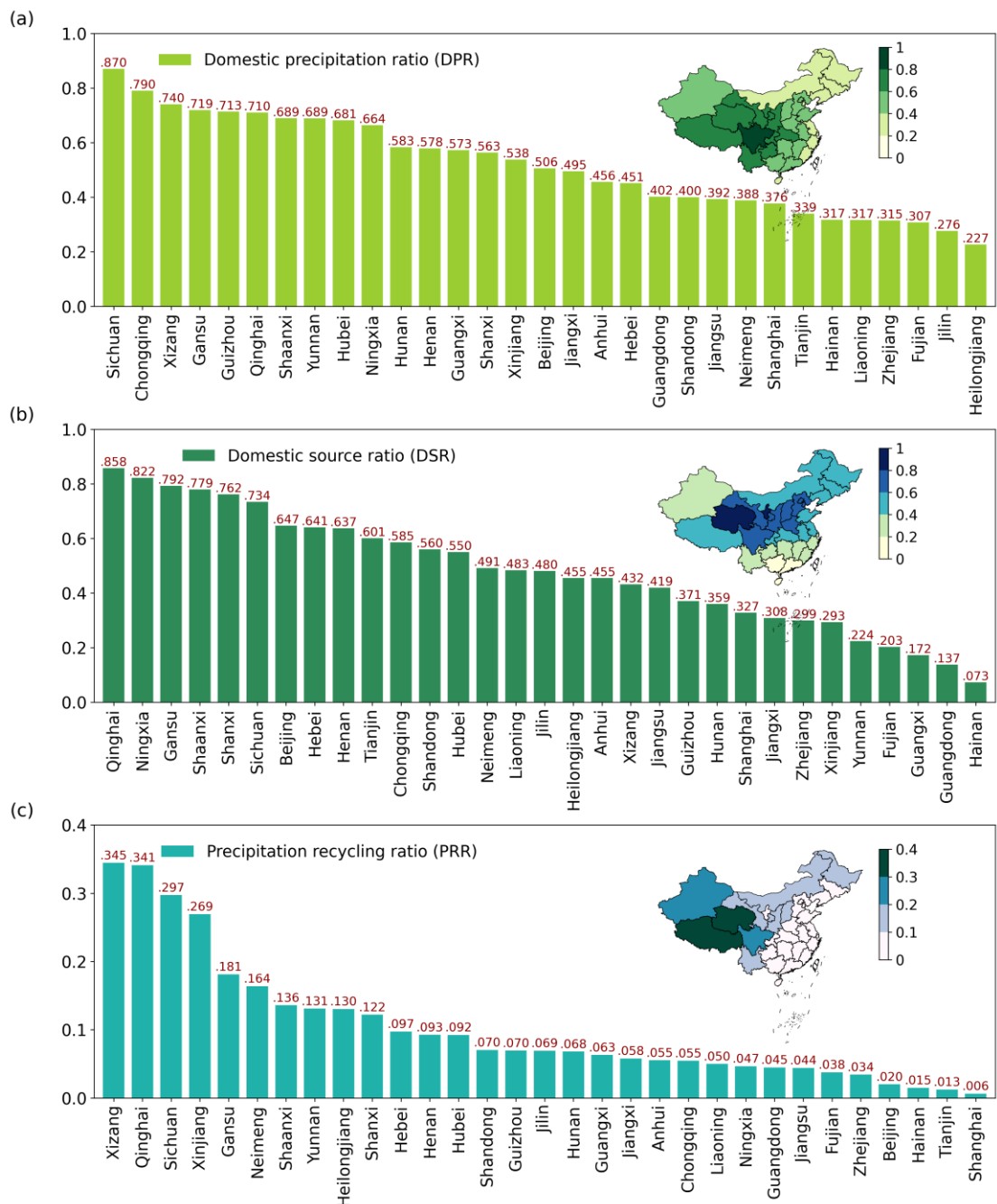

**Figure A2.** (a) Domestic precipitation ratio (DPR), (b) domestic source ratio (DSR) and (c) precipitation recycling ratio (PRR) in each province.

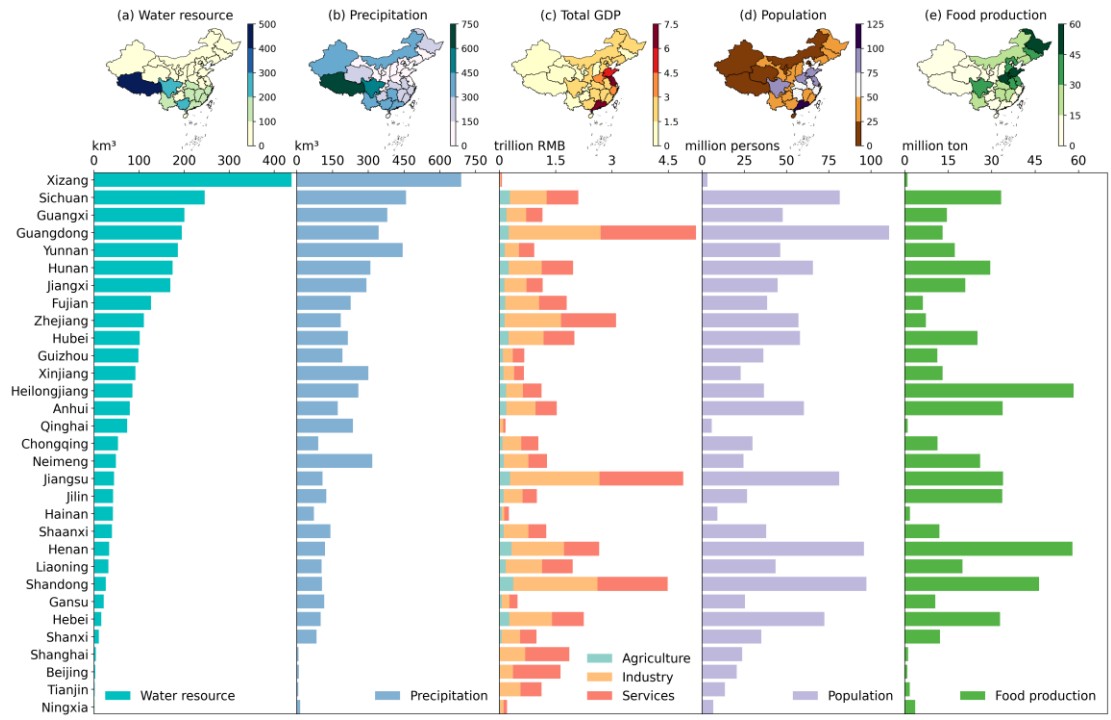

**Figure A3.** Water resource (a), precipitation (b), GDP (c), population (d) and food production (e) in each province.

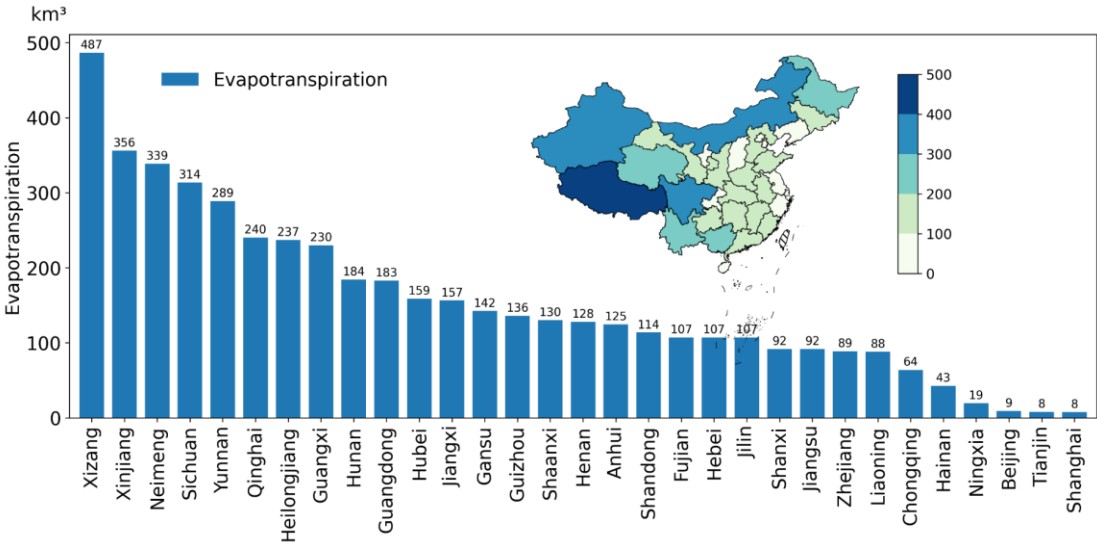

**Figure A4.** Mean evapotranspiration of 2008 to 2017 in each province.

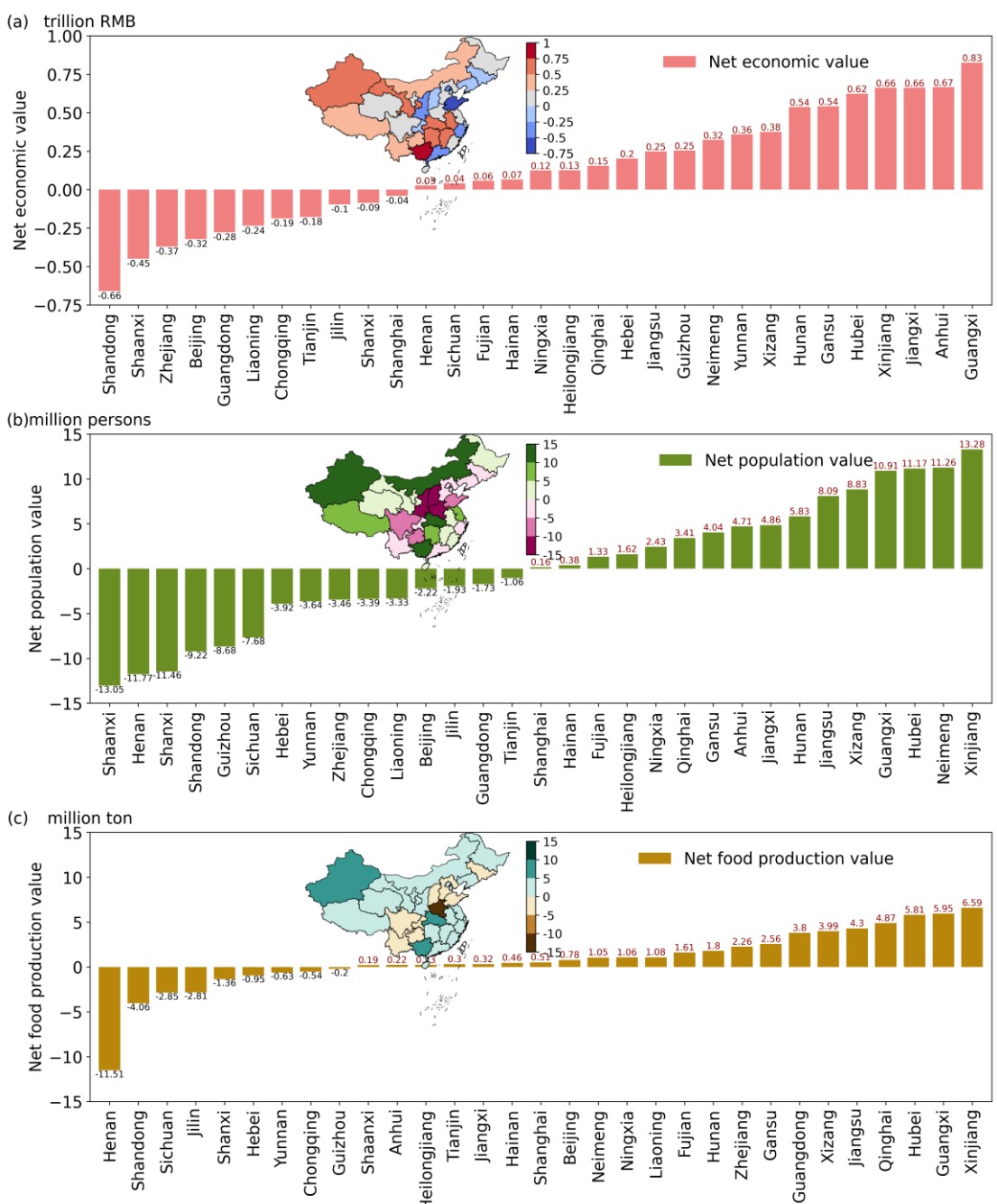

**Figure A5.** Net GDP (a), population (b), food production (c) value of green water flow in each
source province to sink provinces. Positive values represent these socio-economic values of water
resource formed by green water increase by flowing from source to sink provinces. Negative
values represent these socio-economic values of water resource formed by green water decrease
by flowing.

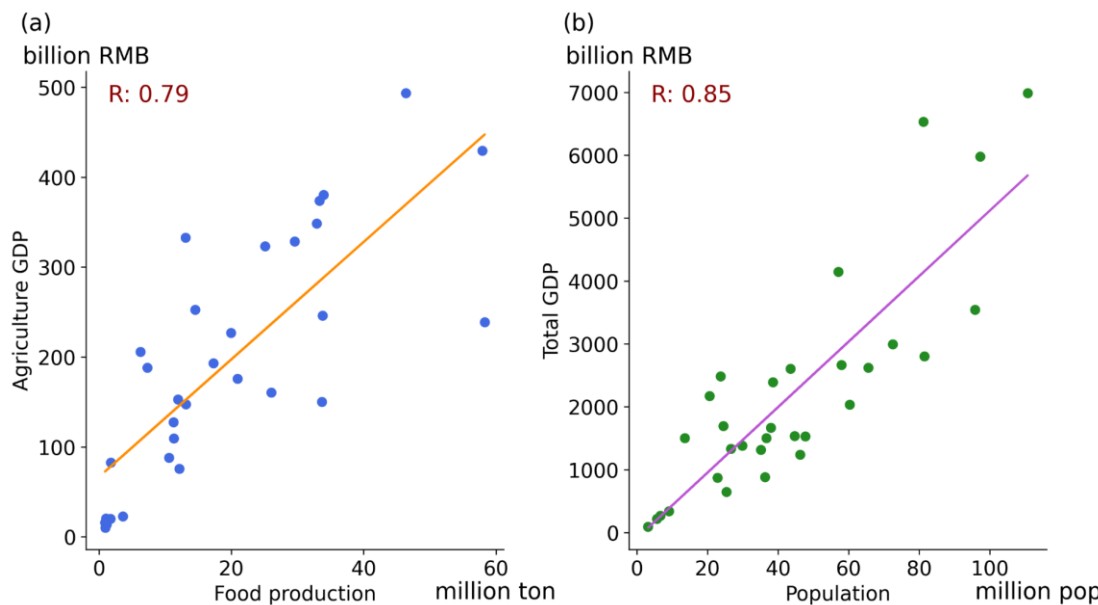


**Figure A6.** Spatial pearson correlation coefficient between agricultural GDP and food production
(a), population and total GDP (b) across provinces in China.

**Table A1.** Precipitation, water resources, and the contribution from green water in provinces of
China.

| Province | Local precipitation (km³) | Precipitation formed by green water (km³) | Percentage of precipitation contribution to local precipitation (%) | Local water resource (km³) | Water resource formed by green water (km³) | Percentage of water resource contribution to local water resource (%) |
|---|---|---|---|---|---|---|
| Beijing | 9.47 | 4.53 | 48 | 2.82 | 1.14 | 40 |
| Tianjin | 7.12 | 2.66 | 37 | 1.62 | 0.70 | 43 |
| Hebei | 100.50 | 48.35 | 48 | 15.98 | 12.26 | 77 |
| Shanxi | 82.88 | 51.69 | 62 | 10.91 | 12.38 | 113 |
| Neimeng | 317.11 | 131.57 | 41 | 48.79 | 31.80 | 65 |
| Liaoning | 104.53 | 27.80 | 27 | 31.92 | 8.40 | 26 |
| Jilin | 124.15 | 29.55 | 24 | 42.21 | 8.98 | 21 |
| Heilongjiang | 258.88 | 53.75 | 21 | 85.40 | 15.44 | 18 |
| Shanghai | 8.02 | 2.83 | 35 | 4.04 | 1.19 | 29 |
| Jiangsu | 108.09 | 35.93 | 33 | 44.27 | 13.43 | 30 |
| Zhejiang | 184.72 | 27.98 | 15 | 110.66 | 13.46 | 12 |
| Anhui | 172.36 | 56.84 | 33 | 79.67 | 23.19 | 29 |
| Fujian | 226.74 | 32.96 | 15 | 126.39 | 17.33 | 14 |
| Jiangxi | 292.56 | 77.52 | 26 | 169.44 | 39.25 | 23 |
| Shandong | 105.99 | 45.49 | 43 | 25.99 | 13.56 | 52 |
| Henan | 118.83 | 73.87 | 62 | 33.73 | 24.08 | 71 |
| Hubei | 214.46 | 108.13 | 50 | 101.66 | 45.27 | 45 |

| Hunan | 308.87 | 107.25 | 35 | 174.33 | 52.28 | 30 |
| Guangdong | 344.05 | 73.31 | 21 | 194.77 | 38.54 | 20 |
| Guangxi | 379.82 | 131.63 | 35 | 200.76 | 66.32 | 33 |
| Hainan | 72.47 | 13.50 | 19 | 41.86 | 7.13 | 17 |
| Chongqing | 90.61 | 50.45 | 56 | 53.23 | 21.87 | 41 |
| Sichuan | 458.97 | 272.93 | 59 | 245.86 | 124.43 | 51 |
| Guizhou | 191.84 | 97.05 | 51 | 98.49 | 46.54 | 47 |
| Yunnan | 444.68 | 199.06 | 45 | 185.99 | 96.34 | 52 |
| Xizang | 689.68 | 360.21 | 52 | 438.59 | 200.33 | 46 |
| Shaanxi | 141.21 | 89.70 | 64 | 39.82 | 26.14 | 66 |
| Gansu | 115.45 | 102.36 | 89 | 21.60 | 30.31 | 140 |
| Qinghai | 236.12 | 170.62 | 72 | 73.50 | 63.57 | 86 |
| Ningxia | 14.95 | 12.94 | 87 | 0.98 | 3.34 | 342 |
| Xinjiang | 300.10 | 191.37 | 64 | 91.95 | 64.92 | 71 |
| Total | 6225.19 | 2683.84 | 43 | 2.82 | 1.14 | 40 |


**Table A2.** The embodied socio-economic values of green water flow from source
provinces for water resources, GDP by industry, population, and food production.
Socio-economic indictors are the average value of 2008-2017.

| Province | Total GDP (Trillion CNY) | Agriculture GDP (Trillion CNY) | Industry GDP (Trillion CNY) | Service GDP (Trillion CNY) | Population (Million persons) | Food production (Million ton) |
|---|---|---|---|---|---|---|
| Beijing | 0.13 | 0.01 | 0.05 | 0.07 | 2.05 | 0.97 |
| Tianjin | 0.09 | 0.01 | 0.04 | 0.04 | 1.33 | 0.61 |
| Hebei | 1.27 | 0.09 | 0.56 | 0.62 | 22 | 10.82 |
| Shanxi | 1.18 | 0.09 | 0.54 | 0.55 | 22.36 | 10.35 |
| Neimeng | 1.67 | 0.15 | 0.77 | 0.75 | 30.77 | 21.78 |
| Liaoning | 0.40 | 0.04 | 0.19 | 0.17 | 7.23 | 5.92 |
| Jilin | 0.27 | 0.03 | 0.12 | 0.11 | 5.34 | 6.37 |
| Heilongjiang | 0.39 | 0.05 | 0.17 | 0.17 | 8.04 | 10.45 |
| Shanghai | 0.09 | 0.01 | 0.04 | 0.04 | 1.41 | 0.57 |
| Jiangsu | 1.06 | 0.08 | 0.51 | 0.47 | 18.13 | 8.5 |
| Zhejiang | 0.69 | 0.04 | 0.32 | 0.32 | 11.08 | 4.11 |
| Anhui | 1.37 | 0.11 | 0.66 | 0.59 | 25.42 | 11.85 |
| Fujian | 0.56 | 0.04 | 0.27 | 0.26 | 9.46 | 2.93 |
| Jiangxi | 1.33 | 0.10 | 0.64 | 0.59 | 24.34 | 9.43 |
| Shandong | 1.37 | 0.11 | 0.65 | 0.62 | 23.85 | 11.72 |
| Henan | 1.75 | 0.15 | 0.85 | 0.75 | 34.94 | 16.74 |
| Hubei | 1.98 | 0.18 | 0.96 | 0.84 | 40.57 | 18.56 |
| Hunan | 1.63 | 0.15 | 0.78 | 0.70 | 33.2 | 14.13 |

| | | | | | | |
|---|---|---|---|---|---|---|
| Guangdong | 1.10 | 0.08 | 0.52 | 0.49 | 20.09 | 6.38 |
| Guangxi | 1.54 | 0.14 | 0.73 | 0.67 | 33.06 | 12.7 |
| Hainan | 0.18 | 0.01 | 0.08 | 0.08 | 3.42 | 1.05 |
| Chongqing | 0.66 | 0.06 | 0.32 | 0.28 | 14.92 | 6.4 |
| Sichuan | 2.31 | 0.25 | 1.10 | 0.96 | 58.39 | 24.16 |
| Guizhou | 1.08 | 0.11 | 0.51 | 0.46 | 25.05 | 10.25 |
| Yunnan | 1.48 | 0.16 | 0.69 | 0.63 | 38.21 | 14.98 |
| Xizang | 0.56 | 0.07 | 0.26 | 0.23 | 15.32 | 5.97 |
| Shaanxi | 1.48 | 0.13 | 0.71 | 0.64 | 30.87 | 14 |
| Gansu | 1.05 | 0.10 | 0.50 | 0.46 | 24.22 | 10.96 |
| Qinghai | 0.72 | 0.07 | 0.34 | 0.31 | 18.3 | 7.56 |
| Ningxia | 0.18 | 0.02 | 0.09 | 0.08 | 3.88 | 1.85 |
| Xinjiang | 1.00 | 0.11 | 0.46 | 0.43 | 22.03 | 11.6 |
| Total (percentage of total contribution to local socio-economic value) | 30.56 (45%) | 2.74 (46%) | 14.43 (45%) | 13.39 (44%) | 629.28 (46%) | 293.67 (50%) |
