# Peer review of "The interprovincial green water flow in China and its tele-connected effects on socio-economy"

_EGUsphere, 2024_

## Referee Comment (RC3)

Review of the manuscript:

**The interprovincial green water flow in China and its tele-connected effects on socio-economy**

**Comments**

The present manuscript provides an emblematic example of integrating green water flows at a sub-national level in water management strategies. It expands on recent studies that highlighted the socio-economic value of green water teleconnections. The topic is suitable for publication and of interest to the readership of EGUsphere.

I would recommend the publication of this paper after major revisions. In the following, there are some comments that the authors may want to consider when revising their manuscript. These revisions should enhance the manuscript's clarity and depth.

**Specific comments:**

**Abstract:**

Lines 15-16: Pay attention to verb consistency for better clarity and flow.
Lines 19-21: The dataset used for the analysis is not well introduced or explained. Provide a more detailed and concise explanation of the data used.
Line 22: Include the specific value of the average self-recycling ratio.
Lines 32-35: This passage is unclear. Consider rephrasing and supporting it with specific results.

**Introduction:**

Line 45: Consider adding additional references for the average global terrestrial moisture recycling ratio. Rockström (2023) cites Tuinenburg (2020), they are essentially the same reference.

Lines 60-61: Clarify the period of reference for the change mentioned. Specify when the change occurred and add recent references for support.

Lines 99-103: This section is unclear. Rephrase for better clarity.

General Comments:

The Introduction could benefit from clearer explanations of certain passages. Include a characterization of China's moisture recycling patterns, atmospheric circulation, and climatic seasonality to frame the phenomenon of moisture flows. For instance, compare the importance of moisture recycling in China to other regions globally.

Discuss the socio-economic background of the Chinese provinces involved. Highlight key socio-economic sectors and societal issues/characteristics of these regions.

Explain why analyzing green water flows at an inter-regional scale is significant, both generally and specifically for China.

**Data and Methods:**

General Comment: This section requires substantial improvements.

Structure: Separate the Data and Methods into two subsections. Move Figure 1 to the Methods subsection and provide detailed explanations in the caption.

Figure 1: The caption should be more detailed to enhance understanding.

Lines 127-138: Provide a more detailed explanation of the reconstruction of flows from the UTrack dataset. Clarify the processing with zonal statistics, possibly using equations or schemes for better comprehension.

Socio-economic Analysis: Since this is the core of the study, it needs a more in-depth analysis. Explain the significance of green water flows for the variables considered. How do they contribute to the services these variables represent?

Equation 1:

- Consider incorporating the areal extension of the provinces. Using population density instead of population, and expressing surface water resources per unit area, would be more appropriate. Similarly, express food production per area rather than gross food production. Use GDP per capita (GDP/P) instead of gross GDP.
- Address the role of irrigation and irrigation infrastructure in food production to avoid overestimating the contribution of green water flows. Differentiating between irrigated and rainfed productivity could be insightful.
- Include units of measure.

Equations 2 and 3:

The focus shifts to the consumption patterns of each province. However, Equation 1 deals with food production, which does not equate to food consumption. Food production in one province might be exported elsewhere. Clarify whether the study focuses on production, consumption, or both, and how these dynamics are analyzed.

**General comment: Consider revising Equations 1,2 and 3 to enhance the rigour of the analysis.**

**Results:**

**Section 3.1:**

Figure 2: The figure has great potential but needs improvements.

- Increase its size for better readability of numbers and histograms.

- Clarify the label of the right bar in the figure, caption, and text. Consider rephrasing for better understanding.

Lines 195-237: The discussion on PRR, DPR, and DSR contains a lot of information that is difficult to visualize. Consider creating a figure to represent these results to help the discussion of socio-economic implications.

Line 214: Provide a definition of westerly winds for a general audience. Also, it is the first time in the manuscript that atmospheric circulation is considered explicitly (see comment about the Introduction)

**Section 3.2:**

Suggest swapping the order with Section 3.1. The geographic visualization of flows in Section 3.2 aids in understanding the results presented in Section 3.1.

**Section 3.3:**

This section is well-written and interesting. However, given its significance to the analysis, consider expanding and providing more in-depth discussion.

**Discussion:**

Overall, this section is well-structured and written, but improvements are needed:

- Enlarge Figure 5 for greater clarity.
- Provide a deeper discussion on the uncertainty of tracked precipitation at the provincial level.
- Lines 429-431: The sentence "Our attempt... [..]" is redundant here and would be more appropriate at the end of the discussion.

**General Comments for Discussion and Conclusions:**

- Include a more in-depth description of the limitations of the socio-economic analysis to add value to these sections and the overall study.
- Discuss potential improvements for this type of socio-economic analysis.

- Since this study is presented as a starting example of integrating green water teleconnections into water management strategies for socio-economic applications, it would be beneficial to elaborate on additional steps needed to achieve this goal.

- Consider discussing other variables that could enhance the analysis of socio-economic implications.

Supplementary Figures and Tables are not cited, and thus integrated in the text. Please integrate them in the main text.

---

## Author Comment (AC1)

We deeply appreciate the detailed and constructive comments provided by the three anonymous reviewers. Following their suggestions and comments, we have extensively revised the manuscript and provided a point-to-point response to each comment. The original comments are in **bold** font, our response is in regular font, and the changes in the text are in blue.

**Comment 1**

**Sang et al. studied the interprovincial connections of green water. They quantified this by calculate the precipitation of each province from the green water inside or outside the province. The work is based on the data generated from a previous particle tracking work. Authors connect the results with social-economic effects, which is very interesting and novel. The structure and writing of the paper are clear. However, I have some thoughts as follows:**

**Response:** Thank you for taking your time to review our study and provide feedback and comments. In the revision, we added more introductions about the connections between green water and socio-economic indices, and moisture tracking dataset.

1. **I don't think authors have clear enough introduction about how they connect the green water with the social-economic value. The introduction is in lines 140-175, but not clear. There are even no dimensions of the variables, and it is hard to know the relationship between different variables in the equations.**

**Response**:Thank you for the comments.

We apologize for the confusion regarding the introduction about the connections between green water and socio-economic values.

The connection of green water with socio-economic value can be reflected in the cascade that green water from source region forms precipitation in sink region, then precipitation recharges water resources and sustain economic activities, human livelihood and crop growth. We estimated the green water contribution on social-economic values in terms of surface water resource volume, economic output (GDP), population and food production.

In the revision, we added a more detailed introduction about the connections between green water and socio-economic values in Figure 1 and section Method. Please see the revised texts and figure below.

Green water from upwind source provinces flows and precipitates downwind to recharge water resources, and therefore sustains socio-economic activities in sink provinces (Fig. 1). Consequently, precipitation, water resources, and socio-economic factors such as economy activities, human livelihood, and crop growth in sink provinces rely on green water exported from source provinces. Changes in green water may affect water resource volume, and then impacts the water supply for economy activities, livelihood and irrigation of crop growth.

[Figure]

Figure 1. A conceptual diagram depicts the teleconnection of green water flows and their socioeconomic contributions in a cascade manner. Evapotranspiration (green dotted arrows) from source region i flows downwind with prevailing winds (green thick arrows) and precipitates in sink region n, which recharges water sources and sustains socioeconomic activities in sink regions.

2. **It looks like authors assume linear relationships between water and all the social-economic indices. I am not quite sure if this is rigorous. For example, whether the food productivity has the positive, linear relationship with water? Similar question to other social-economic indices.**

**Response:** Thank you for the comments.

We apologize for the confusion regarding the assumption about the relationships between green water and socio-economic indices.

In our assessment, we assume that all socio-economic indices (i.e. water resources volume, economic output, population and food production) are sustained by precipitation originating from green water from source regions. The socio-economic value of ET was calculated using the average values of various socio-economic indicators from 2008 to 2017 to match the climatological moisture trajectories from 2008 to 2017. Any temporal changes in water use efficiency per socio-economic variables are not included in this calculation. Therefore, the positive or negative relationship between socio-economic indicators and water use through time does not affect the value assessment of green water.

**3. For the sections of sources and sinks of green water (sections 3.1 and 3.2), it is hard to say they are really novel as it looks like some known results with a new wrapper. You are talking about the evapotranspiration circulation by adding the 'interprovincial' concept.**

**Response:** Thank you for the comments.

Sections 3.1 and 3.2 present the results of interprovincial green water transfer. Although many research analyzed the spatial pattern of moisture recycling in China from amphoteric and hydrological sciences, they identified moisture source and sinks at the grid (Zhang et al., 2023), river basins (Wang et al., 2023), and ecological regions scale (Xie et al., 2024). There is a clear understanding of the large-scale spatial pattern of moisture circulation, few studies focus on quantify moisture recycling at the administrative district scales, which is important for the water management. Therefore, this study applies moisture recycling techniques to inform green water transfer at among provinces, which is previously less known but important for regional water resources management.

With this purpose, Sections 3.1 and 3.2 can help water resources managers understand the moisture recycling and the network of interprovincial moisture recycling is the basis of the socio-economic contribution of green water in each province.

**4. Authors said the data are high quality data from previous studies. I think a bit more introduction is necessary.**

**Response:** Thank you for the comments.

In the revision, we added a more detailed introduction about the UTrack moisture tracking dataset. The revised texts are shown below:

This study used the moisture trajectory dataset generated by the Lagrangian moisture tracking model "UTrack-atmospheric-moisture" driven by ERA5 reanalysis data. The model is the state-of-the-art moisture tracking model, producing more detailed evaporation footprints due to the highest spatial resolution and reducing unnecessary complexity (Tuinenburg and Staal, 2020). The dataset provides monthly moisture flows at the global scale with a spatial resolution of 0.5° for 2008-2017, expressed as the fractions of evaporation from a source grid allocated to precipitation at a sink grid (Tuinenburg et al., 2020). It has been widely used in moisture recycling research with various spatial scales, such as precipitation source of grid (Staal et al., 2023; Wei et al., 2024; Zhang et al., 2023) and basin scale (Wang et al., 2023), and moisture transport between nations (Dirmeyer et al., 2009).

5. **It is a long-term dataset. So why not analyze the temporal variations of these teleconnections? I think this is more important to audience. The average state is also important, but they are the natural pattern which are caused by the long-term climatic conditions. We should know this basic pattern, but we cannot change it much. The more important thing is the temporal variations which represent the variations caused by some interannual variations of natural conditions or by man-made climate change. This is important to inform the future social-economic development, e.g., if such variations are good or bad and if we need actions to control or facilitate such variations.**

**Response**:Thank you for the comments.

We totally agree with that temporal variations of these teleconnections are very important for the social-economic development and water management. However, the available UTrack moisture tracking dataset only provides multi-year monthly mean state of moisture flow for 2008–2017, preventing us from analyzing interannual variations in moisture trajectory caused by natural or man-made factors. We hope analyze interannual variations of green water teleconnections in subsequent research.

In the revision, we added this point in the section Discussion and the revised texts are shown below.

Moreover, the resulting moisture trajectory data only represent the climatologically average moisture trajectories and ET (Li et al., 2023), neglecting the interannual variability in moisture flow trajectory, e.g., those induced by the influence of extreme weather events or ENSO (Zhao and Zhou, 2021). The interannual variations in green

water flow may affect DPR and DSR in some provinces. Human adaptation tends to buffer impacts of interannual variations on socio-economy through water resource management such as reservoirs, dams and other infrastructure. Accounting for interannual variations in green water flows and their socio-economic contribution is worthy future investigation.

**Reference:**

Dirmeyer, P. A., Brubaker, K. L., and DelSole, T.: Import and export of atmospheric water vapor between nations, Journal of Hydrology, 365, 11–22, https://doi.org/10.1016/j.jhydrol.2008.11.016, 2009.

Li, Y., Xu, R., Yang, K., Liu, Y., Wang, S., Zhou, S., Yang, Z., Feng, X., He, C., Xu, Z., and Zhao, W.: Contribution of Tibetan Plateau ecosystems to local and remote precipitation through moisture recycling, Global Change Biology, 29, 702–718, https://doi.org/10.1111/gcb.16495, 2023.

Staal, A., Koren, G., Tejada, G., and Gatti, L. V.: Moisture origins of the Amazon carbon source region, Environ. Res. Lett., 18, 044027, https://doi.org/10.1088/1748-9326/acc676, 2023.

Tuinenburg, O. A. and Staal, A.: Tracking the global flows of atmospheric moisture and associated uncertainties, Hydrology and Earth System Sciences, 24, 2419–2435, https://doi.org/10.5194/hess-24-2419-2020, 2020.

Tuinenburg, O. A., Theeuwen, J. J. E., and Staal, A.: High-resolution global atmospheric moisture connections from evaporation to precipitation, Earth System Science Data, 12, 3177–3188, https://doi.org/10.5194/essd-12-3177-2020, 2020.

Wang, Y., Liu, X., Zhang, D., and Bai, P.: Tracking Moisture Sources of Precipitation Over China, Journal of Geophysical Research: Atmospheres, 128, e2023JD039106, https://doi.org/10.1029/2023JD039106, 2023.

Wei, F., Wang, S., Fu, B., Li, Y., Huang, Y., Zhang, W., and Fensholt, R.: Quantifying the precipitation supply of China's drylands through moisture recycling, Agricultural and Forest Meteorology, 352, 110034, https://doi.org/10.1016/j.agrformet.2024.110034, 2024.

Xie, D., Zhang, Y., Zhang, M., Tian, Y., Cao, Y., Mei, Y., Liu, S., and Zhong, D.: Hydrological impacts of vegetation cover change in China through terrestrial moisture recycling, Science of The Total Environment, 915, 170015, https://doi.org/10.1016/j.scitotenv.2024.170015, 2024.

Zhang, B., Gao, H., and Wei, J.: Identifying potential hotspots for atmospheric water

resource management and source-sink analysis, CSB, 68, 2678–2689, https://doi.org/10.1360/TB-2022-1275, 2023.

Zhao, Y. and Zhou, T.: Interannual Variability of Precipitation Recycle Ratio Over the Tibetan Plateau, Journal of Geophysical Research: Atmospheres, 126, e2020JD033733, https://doi.org/10.1029/2020JD033733, 2021.

---

## Author Comment (AC2)

We deeply appreciate the detailed and constructive comments provided by the three anonymous reviewers. Following their suggestions and comments, we have extensively revised the manuscript and provided a point-to-point response to each comment. The original comments are in **bold** font, our response is in regular font, and the changes in the text are in blue.

**Comment 2**

**The paper presents a novel and insightful approach to connect green water flows to their economic value across different provinces in China. The manuscript is well-structured and written. However, there are a few points that could be clarified and improved to enhance the overall impact and clarity of the study.**

**Response:** Thank you for taking your time to review our study and provide feedback and comments. In the revision, we added more introductions about the connections between green water and socio-economic indices.

1  **The use of the UTrack dataset for processing and tracking the green water flow is commendable. But the uncertainties associated with the input precipitation data (P) in the forward-tracking process need further elaboration.**

    1.1  **It would be good to include the processing scripts for the UTrack dataset and a clearer explanation how you got from the dataset to the data that is used in the already included Python notebook.**

    1.2  **Additionally, this preprint (https://www.researchsquare.com/article/rs-4177311/v1) indicates that the water balance in the UTrack dataset does not appear to be closed, which leads to under- and overestimations of P and ET in either tracking direction. While this may be difficult to solve in the scope of this study, its implications should be discussed and made aware of. How could such uncertainties be addressed and how do they influence the results of this study?**

**Response:** Thank you for the comments.

1. Uncertainties associated with the input precipitation data (P) affect the precipitation contribution and precipitation recycling ratio.

The revised texts in section Discussion are shown below:

First, ET and precipitation datasets driving the UTrack model affect the tracked trajectories and magnitude of moisture flow. The resulting moisture trajectory is expressed as the fraction of ET to precipitation, and the amount of moisture is restored by the ET and precipitation dataset chosen by users. Different ET and precipitation datasets lead to different precipitation contribution and PRR (Li et al., 2023). We used the ERA5 dataset to keep consistent with the original UTrack model. It is noted that the non-closure of the moisture balance from ERA5 (Tuinenburg et al., 2020) results in more uncertainties in moisture tracking results (De Petrillo et al., 2024).

1.1 We will share all the processing scripts for the UTrack dataset and our analysis before publication of the paper.

1.2 The preprint by De Petrillo.et al indicates deviations between the tracked evaporation (ET) and precipitation (P) volumes at the country/ocean scale, because of the unclosed water balance by ERA5. The non-closure of water balance leads to more uncertainties in capturing moisture tracking. This preprint used the Iterative Proportional Fitting (IPF) procedure to adjust the tracked moisture flow matrices to ensure that the total evaporation and precipitation volumes match the reanalysis data at the country/ocean scale.

We estimated the ratio of P in sink province originating from ET of different source province, and then used the observed provincial P to estimate the P contribution from each province. This practice ensures that provincial P is 100% decomposed into different sources, avoiding the estimation bias of sink precipitation due to unclosed water balance by ET and P data.

**2. While the paper shows a novel approach of linking green water flows to economic values, the connection between green water and socio-economic outcomes needs more clarity.**

    **2.1 Green water appears to be treated similarly to blue water under the assumption that all green water flows can be used by humans and directly transferred into an economic value. But large parts of the flows probably remain inaccessible for direct human use and are rather important for indirect ecosystem services, and the stability of the carbon and hydrologic cycle.**

    **2.2 More clarity is needed to understand how the link between the green**

**water flows and the socio-economic values are made. This is the novel part of this paper and would benefit from making this connection clearer. For instance, sections 3.1 and 3.2 present results which aren't really novel but rather an exploration under a different geographical lens. Section 3.3 presents the novel results of this study and hence, by streamlining 3.1 and 3.2 space could be made to focus and expand on the connection between green water and socio-economic values in China.**

**Response:** Thank you for the comments.

2. We apologize for the confusion regarding the introduction about the connections between green water and socio-economic values.

The tele-connection of green water and socio-economic value is depicted as cascade: green water forms precipitation, then precipitation recharges water resource, which then sustains economic activities, human livelihood and crop growth. So we used the water resource volume, economic output (GDP), population and food production, the four variables, to estimate the green water contribution.

In the revision, we clarify these in section 2.2 and Figure 1. Please see the revised texts and figure below.

Green water from upwind source provinces flows and precipitates downwind to recharge water resources, and therefore sustains socio-economic activities in sink provinces (Fig. 1). Consequently, precipitation, water resources, and socio-economic factors such as economy activities, human livelihood, and crop growth in sink provinces rely on green water exported from source provinces. Changes in green water may affect water resource volume, and then impacts the water supply for economy activities, livelihood and irrigation of crop growth. We choose water resources volume, economy output (GDP), population and food production, the four socio-economic indictors that are tightly related to water resources (Li et al., 2018) to evaluate green water socio-economic contributions of the source province and capture the social dynamics of moisture recycling (Keys and Wang-Erlandsson, 2018).

[Figure]

Figure 1. A conceptual diagram depicts the teleconnection of green water flows and their socioeconomic contributions in a cascade manner. Evapotranspiration (green dotted arrows) from source region i flows downwind with prevailing winds (green thick arrows) and precipitates in sink region n, which recharges water sources and sustains socioeconomic activities in sink regions.

2.1 We totally agree with that large parts of green water flows probably remain inaccessible for direct human use. Only the water resources generated from precipitation and consumed by human activities sustains the socio-economy. We used the ratio of water resources formed by precipitation (P) in the calculation of green water value. Since this ratio can be eliminated during the calculation process, Equation (1) is simplified and does not show the ratio involved in the calculation. The process of simplification is below. We take the GDP value of ET in one province as an example:

$$water\ resouces\ value\ of\ green\ water$$
$$= \frac{water\ resources}{P} \times P \times green\ water\ forming\ P\ fraction$$
$$= water\ resources \times green\ water\ forming\ P\ fraction$$

$GDP\ value\ of\ green\ water$

$$= \frac{water\ resources\ value\ of\ green\ water}{water\ resources} \times GDP\ value$$

$$= \frac{\cancel{water\ resources}\ \times green\ water\ forming\ P\ fraction}{\cancel{water\ resources}}$$

$$\times GDP\ vlalue = green\ water\ forming\ P\ fraction\ \times GDP\ value$$

This research assume that all the used water resources are from precipitation generated by green water. This point was added in the Discussion and revised texts are shown below:

Secondly, the socio-economic value assessment of green water in this study only considers green water flows within China, excluding flows moving abroad and to the ocean that may embody socio-economic value beyond territory of mainland China. We mainly attribute socio-economic values to green water and generated precipitation, because precipitation is the ultimate water source to recharge surface and groundwater of a region. Strictly speaking, such attribution is not precise because socio-economy also utilizes streamflow from upstream areas which deserve separate attention. In addition, not all water resources replenished by green water-generated precipitation are accessible for human activities, part of them serve the nature ecosystem, such as sustaining forest function (Keys et al., 2019). Despite three socio-economic indices closely linked to water resources, the socio-economic contribution of green water flows can be manifested in other aspects such as livestock production, irrigated agriculture. In future study, green water's socio-economic contribution can be assessed annually by using moisture tracking dataset containing interannual variations and with or separation of water resources consumed by socio-economy (surface and ground water), which is more conducive to understand long-term changes and variations in the linkage between green water, water resources and economic development.

2.2 Sections 3.1 and 3.2 show the results of interprovincial green water transfer. Many studies have analyzed the pattern of moisture recycling in China from amphoteric and hydrological sciences, they identified moisture source and sinks at the grid (Zhang et al., 2023), basin (Wang et al., 2023) and ecological regional scale (Xie et al., 2024). These studies present a clear understanding of the large-scale spatial pattern of moisture circulation. Few studies focus on the moisture recycling analysis at the administrative district scale, which is important for the water management. Therefore, this study promotes the application of moisture recycling at the water resources management scale, which is also important for research outside the field of atmospheric science.

Sections 3.1 and 3.2 can help water resources managers understand the moisture recycling. The network of interprovincial moisture recycling is the basis of the socio-economic contribution of green water in each province.

In the revision, we will streamline 3.1 and 3.2.

**Reference:**

De Petrillo, E., Fahrländer, S., Tuninetti, M., Andersen, L. S., Monaco, L., Ridolfi, L., and Laio, F.: Reconciling tracked atmospheric water flows to close the global freshwater cycle, https://doi.org/10.21203/rs.3.rs-4177311/v1, 30 April 2024.

Keys, P. W. and Wang-Erlandsson, L.: On the social dynamics of moisture recycling, Earth System Dynamics, 9, 829–847, https://doi.org/10.5194/esd-9-829-2018, 2018.

Keys, P. W., Porkka, M., Wang-Erlandsson, L., Fetzer, I., Gleeson, T., and Gordon, L. J.: Invisible water security: Moisture recycling and water resilience, Water Security, 8, 100046, https://doi.org/10.1016/j.wasec.2019.100046, 2019.

Li, X., Zhang, Q., Liu, Y., Song, J., and Wu, F.: Modeling social–economic water cycling and the water–land nexus: A framework and an application, Ecological Modelling, 390, 40–50, https://doi.org/10.1016/j.ecolmodel.2018.10.016, 2018.

Li, Y., Xu, R., Yang, K., Liu, Y., Wang, S., Zhou, S., Yang, Z., Feng, X., He, C., Xu, Z., and Zhao, W.: Contribution of Tibetan Plateau ecosystems to local and remote precipitation through moisture recycling, Global Change Biology, 29, 702–718, https://doi.org/10.1111/gcb.16495, 2023.

Tuinenburg, O. A., Theeuwen, J. J. E., and Staal, A.: High-resolution global atmospheric moisture connections from evaporation to precipitation, Earth System Science Data, 12, 3177–3188, https://doi.org/10.5194/essd-12-3177-2020, 2020.

Wang, Y., Liu, X., Zhang, D., and Bai, P.: Tracking Moisture Sources of Precipitation Over China, Journal of Geophysical Research: Atmospheres, 128, e2023JD039106, https://doi.org/10.1029/2023JD039106, 2023.

Xie, D., Zhang, Y., Zhang, M., Tian, Y., Cao, Y., Mei, Y., Liu, S., and Zhong, D.: Hydrological impacts of vegetation cover change in China through terrestrial moisture recycling, Science of The Total Environment, 915, 170015, https://doi.org/10.1016/j.scitotenv.2024.170015, 2024.

Zhang, B., Gao, H., and Wei, J.: Identifying potential hotspots for atmospheric water resource management and source-sink analysis, CSB, 68, 2678–2689, https://doi.org/10.1360/TB-2022-1275, 2023.

---

## Author Comment (AC3)

We deeply appreciate the detailed and constructive comments provided by the three anonymous reviewers. Following their suggestions and comments, we have extensively revised the manuscript and provided a point-to-point response to each comment. The original comments are in **bold** font, our response is in regular font, and the changes in the text are in blue.

**Comment 3**

**The present manuscript provides an emblematic example of integrating green water flows at a sub-national level in water management strategies. It expands on recent studies that highlighted the socio-economic value of green water teleconnections. The topic is suitable for publication and of interest to the readership of EGUsphere.**

**Response:** Thank you for taking your time to review our study and provide feedback and comments.

**I would recommend the publication of this paper after major revisions. In the following, there are some comments that the authors may want to consider when revising their manuscript. These revisions should enhance the manuscript's clarity and depth.**

**Specific comments:**

**Abstract:**

**Lines 15-16: Pay attention to verb consistency for better clarity and flow.**

**Response:** Thank you for the comments.

We modified this part to be more specific, and the revised texts are shown below:

Green water (terrestrial evapotranspiration), flows from source regions and precipitates downwind via moisture recycling, recharges water resources and sustains socio-economy in sink regions.

**Lines 19-21: The dataset used for the analysis is not well introduced or explained. Provide a more detailed and concise explanation of the data used.**

**Response:** Thank you for the comments.

In the revision, we revised the sentence to include more information. The revised texts

are shown below:

This study used the moisture tracking dataset which contains monthly moisture flows at the global scale with a spatial resolution of 0.5° for 2008-2017 to quantify interprovincial green water flows in China and their socio-economic contributions.

And in Section 2.1, we added a more detailed introduction about the UTrack moisture tracking dataset. The revised texts are shown below:

This study used the moisture trajectory dataset generated by the Lagrangian moisture tracking model "UTrack-atmospheric-moisture" driven by ERA5 reanalysis data. The model is the state-of-the-art moisture tracking model, producing more detailed evaporation footprints due to the highest spatial resolution and reducing unnecessary complexity (Tuinenburg and Staal, 2020). The dataset provides monthly moisture flows at the global scale with a spatial resolution of 0.5° for 2008-2017, expressed as the fractions of evaporation from a source grid allocated to precipitation at a sink grid (Tuinenburg et al., 2020). It has been widely used in moisture recycling research with various spatial scales, such as precipitation source of grid (Staal et al., 2023; Wei et al., 2024; Zhang et al., 2023) and basin scale (Wang et al., 2023), and moisture transport between nations (Dirmeyer et al., 2009).

**Line 22: Include the specific value of the average self-recycling ratio.**

**Response:** Thank you for the comments.

In the revision, we added the range of self-recycling ratio, and revised texts are shown below:

The precipitation recycling rate (PRR) range of 31 provinces is from 0.6% to 35%.

**Lines 32-35: This passage is unclear. Consider rephrasing and supporting it with specific results.**

**Response:** Thank you for the comments.

In the revision, we rephrased this passage, and the revised texts are shown below:

Green water primarily flows from underdeveloped regions to developed provinces, such as Xizang and Xinjiang whose exported green water having the most added socio-economic value. This suggests that less developed provinces effectively support the higher socio-economic status of developed provinces through green water supply.

**Introduction:**

**Line 45: Consider adding additional references for the average global terrestrial moisture recycling ratio. Rockström (2023) cites Tuinenburg (2020), they are essentially the same reference.**

**Response:** Thank you for the comments.

In the revision, we changed the references, and the revised texts are shown below:

Terrestrial evapotranspiration (i.e., green water) (Falkenmark and Rockström, 2006), which includes evaporation and transpiration from land and vegetation, contributes to over half of the global precipitation on land (van der Ent et al., 2010; Theeuwen et al., 2023; Tuinenburg et al., 2020).

**Lines 60-61: Clarify the period of reference for the change mentioned. Specify when the change occurred and add recent references for support.**

**Response:** Thank you for the comments.

In the revision, the period of reference was clarified, and the revised texts are shown below:

global total blue water flow into oceans and internal sinks has decreased by 3.5% in 2002 compared to 1961–1990 due to upstream water withdrawal and dams (Döll et al., 2009)

**Lines 99-103: This section is unclear. Rephrase for better clarity.**

**Response:** Thank you for the comments.

In the revision, we rephrased this passage and revised texts are shown below:

Recent studies analyzed green water flows at the national or regional scale to identify the source and sink areas of specific regions, like the Tibetan Plateau (Zhang et al., 2024) and Europe (Pranindita et al., 2022). However, green water flows from different regions are interlinked and become sources and sinks of each other. Such green water transfer at a sub-national scale effectively forms a complex green water flow network, and highlights the mutual dependency of green water and its socio-economic contributions, especially for large countries like China.

**General Comments:**
**The Introduction could benefit from clearer explanations of certain passages. Include a characterization of China's moisture recycling patterns, atmospheric**

**circulation, and climatic seasonality to frame the phenomenon of moisture flows. For instance, compare the importance of moisture recycling in China to other regions globally.**

**Response:** Thank you for the comments.

In the revision, we added the introduction about China's moisture recycling patterns, atmospheric circulation, and climatic seasonality. The revised texts are shown below:

The moisture flows in China are mainly influenced by prevailing westerly winds (from the west toward the east) in most regions of China between 30° and 60° (Bridges et al., 2023), the East Asian monsoon in eastern China, and India monsoons in southwestern China. In summer, the East Asian summer monsoon and India summer monsoon bring moisture for precipitation in eastern and southwestern China, (Tian and Fan, 2013). In winter, the East Asian winter monsoon drives northwesterly moisture transport across much of China and generate precipitation (Wu and Wang, 2002). These regional atmospheric circulation systems determine the basic spatial and seasonal patterns of green water flow across China.

**Discuss the socio-economic background of the Chinese provinces involved. Highlight key socio-economic sectors and societal issues/characteristics of these regions.**

**Response:** Thank you for the comments.

In the revision, we added the socio-economic background of the Chinese provinces. The revised texts are shown below:

Furthermore, the substantial disparities in socio-economic development among Chinese provinces add complexity to the socio-economic contributions of green water. The western provinces have a weak economic base and sparse populations, but are abundant in water resources (Ya-Feng et al., 2020). The eastern provinces are economically developed and densely populated, whereas they suffer from water scarcity (Varis and Vakkilainen, 2001). There is an urgent need to quantify moisture flows at provincial scale and to evaluate the socio-economic value embedded in interprovincial green water. This offer new perspectives for optimizing water resource utilization and mitigating the imbalance in regional socio-economic development.

**Explain why analyzing green water flows at an inter-regional scale is significant, both generally and specifically for China.**

**Response:** Thank you for the comments.

Although many research analyzed the spatial pattern of moisture recycling in China from amphoteric and hydrological sciences, they identified moisture source and sinks at the grid (Zhang et al., 2023), river basins (Wang et al., 2023), and ecological regions scale (Xie et al., 2024). There is a clear understanding of the large-scale spatial pattern of moisture circulation, few researches focus on quantify moisture recycling at the administrative district scales, which is important for the water management. Therefore, this study applies moisture recycling techniques to inform green water transfer at among provinces, which is previously less known but important for regional water resources management.

In the revision, we will add the explanation.

**Data and Methods:**

**General Comment: This section requires substantial improvements.**

**Structure: Separate the Data and Methods into two subsections. Move Figure 1 to the Methods subsection and provide detailed explanations in the caption.**

**Response:** Thank you for the comments.

In the revision, we separated the Data and Methods into two subsections. And added detailed explanations in the caption of Figure 1.

**Figure 1: The caption should be more detailed to enhance understanding.**

**Response:** Thank you for the comments.

In the revision, we added detailed explanations in the caption of Figure 1.

The revised figure and its caption are shown below:

[Figure]

Figure 1. A conceptual diagram depicts the teleconnection of green water flows and their socioeconomic contributions in a cascade manner. Evapotranspiration (green dotted arrows) from source region i flows downwind with prevailing winds (green thick arrows) and precipitates in sink region n, which recharges water sources and sustains socioeconomic activities in sink regions.

**Lines 127-138: Provide a more detailed explanation of the reconstruction of flows from the UTrack dataset. Clarify the processing with zonal statistics, possibly using equations or schemes for better comprehension.**

**Response:** Thank you for the comments.

In the revision, we added detailed explanations of the reconstruction of flows from the UTrack dataset in Section 2.2 and Figure A1. The revised texts and figure are shown below:

Firstly, we quantified interprovincial moisture flows and their precipitation contribution (Fig. A1). At each sink grid, the ET to precipitation fractions from the moisture trajectory were multiplied by ERA5 evapotranspiration (ET) to obtain monthly precipitation contribution by moisture from all source grids. Repeating the calculation for all grids within a sink province and summing them up yielded the precipitation in sink province contributed by each of source grid (Fig. A1 Step 1). We next employed zonal statistics to sum up source grids by province to obtain precipitation in sink province contributed by each of source province, and the precipitation contribution was

converted to relative values (i.e., the fraction of precipitation in sink province $j$ originating from green water of a source province $i$, denoted as $W_{ij}$) rather than absolute contribution to reduce the uncertainty in the latter (Fig. A1 Step 2). The fractions $W_{ij}$ multiplied by precipitation of sink province restore the absolute precipitation contribution. Finally, the interprovincial green water flows in China was derived after estimating each province individually.

[Figure]

**Figure A1.** Interprovincial green water flowing network calculation. Step 1: calculate the precipitation in sink grids originating from ET in source grids. Step 2: calculate the fraction of precipitation in sink provinces originating from source provinces.

**Socio-economic Analysis: Since this is the core of the study, it needs a more in-depth analysis. Explain the significance of green water flows for the variables considered. How do they contribute to the services these variables represent?**

Response: Thank you for the comments.

In the revision, we added the explanation of green water flows' significance. The revised texts are shown below:

Consequently, precipitation, surface water resources, and socio-economic factors such as economy activities, human livelihood, and crop growth in sink provinces rely on green water exported from source provinces. Green water affects the surface water resource volume and crop growth by precipitating, and then impacts the water supply for economy activities, livelihood and irrigation of crop growth. We choose water resources volume, economy output (GDP), population and food production, the four socio-economic indictors to evaluate green water socio-economic contributions of the source province because they are tightly related to water resources (Li et al., 2018) and can capture the social dynamics of moisture recycling (Keys and Wang-Erlandsson, 2018).

**Equation 1:**

**Consider incorporating the areal extension of the provinces. Using population density instead of population, and expressing surface water resources per unit area, would be more appropriate. Similarly, express food production per area rather than gross food production. Use GDP per capita (GDP/P) instead of gross GDP.**

**Response:** Thank you for the comments.

The land area and population of source and sink provinces differ, when using per-unit measures, the socio-economic value of green water is affected by the denominators of the indicators for both the source and sink provinces, making the results difficult to interpret. Alternatively, per-unit measures reflect the different characteristics between the source and sink regions, such as the green water flowing from economically less developed to developed provinces indicated by per capita GDP. These results have already been discussed and analyzed in the manuscript in Section xxx. Therefore, we choose to use total amounts of socio-economic indicators to calculate the green water value.

**Address the role of irrigation and irrigation infrastructure in food production to avoid overestimating the contribution of green water flows. Differentiating between irrigated and rainfed productivity could be insightful.**

**Response:** Thank you for the comments.

The irrigation increases ET in source region, then provides more moisture for downwind regions. Simultaneously, most of the water for local irrigation comes from upwind regions. Therefore, it is necessary to distinguish water sources more carefully. In the revision, we will add this point in Section Discussion.

However, the food production data from China Statistical Yearbook is the total food production including both irrigated and rainfed production. It's hard to differentiate the irrigated and rainfed productivity due to the data limitation.

**Include units of measure.**

**Response:** Thank you for the comments.

In the revision, we added units and the revised texts are shown below:

Where $S_j$ is the average socio-economic value of 2008-2017 (i.e., surface water resources volume ($km^3$), GDP (CNY, 1 CNY = 0.14 USD), population (persons), and food production (ton)) at sink province $j$, n is the number of sink provinces.

**Equations 2 and 3:**

**The focus shifts to the consumption patterns of each province. However, Equation 1 deals with food production, which does not equate to food consumption. Food production in one province might be exported elsewhere. Clarify whether the study focuses on production, consumption, or both, and how these dynamics are analyzed.**

**Response:** Thank you for the comments.

We apologize for the confusion regarding the research's focus.

Our research focus on the food production. We used water productivity in source province to calculate socio-economic values of its exported green water in the counterfactual scenario when it was all consumed in source province without interprovincial transfer. The results were compared with the actual green water's socio-economic values (namely socio-economic values of exported green water when it is consumed in sink provinces). What we focused is the difference of food production between the scenario that green water is all consumed in source provinces and actual scenario.

In the revision, we will clarify this point more clearly.

**General comment: Consider revising Equations 1,2 and 3 to enhance the rigor of the analysis.**

**Results:**

**Section 3.1:**

**Figure 2: The figure has great potential but needs improvements.**

**Increase its size for better readability of numbers and histograms.**

**Clarify the label of the right bar in the figure, caption, and text. Consider rephrasing for better understanding.**

**Response:** Thank you for the comments.

In the revision, we improved Figure 2 to increase its readability. The revised figure is shown below:

[Figure]

**Lines 195-237: The discussion on PRR, DPR, and DSR contains a lot of information that is difficult to visualize. Consider creating a figure to represent these results to help the discussion of socio-economic implications.**

**Response:** Thank you for the comments.

The information of PRR, DPR, and DSR in each province has shown in Figure A3 (Appendix Figure 3).

[Figure]

**Figure A3.** Domestic precipitation ratio (a), domestic source ratio (b) and precipitation recycling ratio (c) in each province.

**Line 214: Provide a definition of westerly winds for a general audience. Also, it is the first time in the manuscript that atmospheric circulation is considered explicitly (see comment about the Introduction)**

**Response:** Thank you for the comments.

In the revision, we provided the definition of westerly in Introduction. The revised texts are shown below:

The moisture pathways in China is influenced by both midlatitude westerly (prevailing winds from the west toward the east in the middle latitudes between 30° and 60° (Bridges et al., 2023)) and monsoonal circulations, resulting in significant regional variations.

**Section 3.2:**

**Suggest swapping the order with Section 3.1. The geographic visualization of flows in Section 3.2 aids in understanding the results presented in Section 3.1.**

**Response:** Thank you for the comments.

Section 3.1 shows the network of interprovincial green water flows. Section 3.2 shows the spatial pattern of each province's composited green water flows direction based on the green water flowing network. We think it's more appropriate to clarify the interprovincial green water flowing network firstly.

**Section 3.3:**

**This section is well-written and interesting. However, given its significance to the analysis, consider expanding and providing more in-depth discussion.**

**Response:** Thank you for the comments.

The in-depth discussion of Section 3.3 is in the Section Discussion. To make this part clearer, we will move the in-depth discussion to Section Results in the revision.

**Discussion:**

**Overall, this section is well-structured and written, but improvements are needed: Enlarge Figure 5 for greater clarity.**

**Response:** Thank you for the comments.

In the revision, we enlarged Figure 5, and the revised figure is shown below:

[Figure]

**Provide a deeper discussion on the uncertainty of tracked precipitation at the provincial level.**

**Response:** The uncertainty of tracked precipitation includes three aspects: (1) different ET and precipitation datasets lead to different precipitation contribution and PRR due to the ET amount and spatial distribution; (2)the non-closure of the moisture balance between ET and precipitation from ERA5 results in inaccurately capturing actual ET and precipitation volumes; (3) the moisture tracking model has some assumption and simplification. The purpose of our work is not to precisely quantify the moisture recycling quantification but to reveal the relationship between the moisture and socio-economy.

The revised texts in section Discussion are shown below:

Due to complex dynamics of the green water flow and limitations of the moisture tracking model, there are still major uncertainties in data and methods of this study. First, ET and precipitation datasets driving the UTrack model affect the tracked trajectories of green water flow. The ratio of precipitation formed by ET from the UTrack model has been standardized. So the moisture is determined by the ET and precipitation dataset. Different ET and precipitation datasets lead to different precipitation contribution and PRR due to the ET amount and spatial distribution (Li et al., 2023). We used the ERA5 dataset to keep consistent with the original UTrack model calculation. The non-closure of the moisture balance between ET and precipitation from ERA5 (Tuinenburg et al., 2020) results in inaccurately capturing actual ET and precipitation volumes, affecting water management and policy decisions (De Petrillo et al., 2024). Simplifications and assumptions introduced in the moisture tracking model also add uncertainty (Tuinenburg and Staal, 2020). Moreover, The resulting moisture trajectory data only represent the climatologically average moisture trajectories and ET

(Li et al., 2023), neglecting the interannual variability in green water flow, e.g., those induced by the influence of extreme weather events or ENSO (Zhao and Zhou, 2021). The interannual fluctuations may affect some provinces' DPR and DSR slightly. As for socio-economy, these may have little impact due to the human adaptation, such as reservoirs, dams and other infrastructure.

**Lines 429-431: The sentence "Our attempt... [..]" is redundant here and would be more appropriate at the end of the discussion.**
**Response:** Thank you for the comments.
In the revision, we moved this sentence to the end of the discussion.

**General Comments for Discussion and Conclusions:**
**Include a more in-depth description of the limitations of the socio-economic analysis to add value to these sections and the overall study.**
**Response:** Thank you for the comments.
The limitation of the socio-economic analysis includes:
(1) The green water's socio-economic contribution excludes green water flowing abroad, and did not separate water sources in assessing socio-economic contribution and consider the accessibility of precipitation formed by green water to human activities.
(2) The research selected three socio-economic indices which are tightly linked to the water resources. Other indices linked to water resources like livestock production and irrigated agriculture are not considered.
In the revision, we added these limitations in the section Discussion, and the revised texts are shown below:
Secondly, the socio-economic value assessment of green water in this study only considers green water flows within China, excluding flows moving abroad and to the ocean that may embody socio-economic value beyond territory of mainland China. We mainly attribute socio-economic values to green water and formed precipitation, because precipitation is the ultimate water source of a region (i.e., surface and groundwater). Strictly speaking, such attribution is not precise because socio-economy also utilizes streamflow from upstream areas which deserve separate attention. And the interactions between blue and green water add uncertainties for evaluating the green water's socio-economic values. For example, the irrigation from blue water increases ET in upwind source region, then provides more moisture for downwind sink regions, thus enhancing precipitation by moisture recycling (McDermid et al., 2023). In addition,

water resources from green water flows can't all be accessible for human activities, part of them serve the nature ecosystem, such as sustaining forest function (Keys et al., 2019). The research focused on three socio-economic indices closely linked to water resources to evaluate the value of green water, which overlooked other relevant indices, such as livestock production, irrigated agriculture. So the socio-economic value of green water flows are probably overestimated or underestimate.

**Discuss potential improvements for this type of socio-economic analysis.**

**Response:** Thank you for the comments.

Potential improvements for this type of socio-economic analysis include (1) the moisture recycling dataset improving; (2) the interannual variations analysis of green water's socio-economic contribution; (3) the distinguish of blue and green water source; (4) more comprehensive assessment of the green water's contribution.

In the revision, we added these potential improvements in the section Discussion, and the revised texts are shown below:

In future study, it would be ideal to have more accurate moisture tracking models providing higher precision and spatiotemporal resolution in moisture trajectory data. So that the green water's socio-economic contribution can contain interannual variations, which is more conducive to promoting water resource management and economic development. And the source of water resources sustaining the socio-economy in each province should be distinguished into green and blue water flow to exclude blue water's socio-economic contribution. Moreover, green water's contribution can be evaluated comprehensively through more socio-economic indicators.

**Since this study is presented as a starting example of integrating green water teleconnections into water management strategies for socio-economic applications, it would be beneficial to elaborate on additional steps needed to achieve this goal.**

**Response:** Thank you for the comments.

Additional steps should solve existing deficiencies to improve green water's socio-economic contribution.

In the revision, we added these potential improvements in the section Discussion, and the revised texts are shown below:

In future study, it would be ideal to have more accurate moisture tracking models providing higher precision and spatiotemporal resolution in moisture trajectory data. So that the green water's socio-economic contribution can contain interannual variations,

which is more conducive to promoting water resource management and economic development. And the source of water resources sustaining the socio-economy in each province should be distinguished into green and blue water flow to exclude blue water's socio-economic contribution. Moreover, green water's contribution can be evaluated comprehensively through more socio-economic indicators.

**Consider discussing other variables that could enhance the analysis of socio-economic implications.**

**Response:** Thank you for the comments.

Some other socio-economic variables like livestock and irrigated agriculture may enhance the analysis. In the revision, we added them and the revised texts are shown below:

The research focused on three socio-economic indices closely linked to water resources to evaluate the contribution of green water, which overlooked other relevant indices, such as livestock production, irrigated agriculture.

**Supplementary Figures and Tables are not cited, and thus integrated in the text. Please integrate them in the main text.**

**Response:** Thank you for the comments.

We apologize for the confusion regarding the figures citing and Supplementary Figures and Tables citing. In the main text, supplementary figures were cited as Fig. A1, Fig. A2; supplementary tables were cited as Table A1 and Table A2. All supplementary figures and tables were cited in section Result and Discussion.

**Reference:**

[revised manuscript text omitted]

---

## Author Response (AR1)

We deeply appreciate the detailed and constructive comments provided by the three anonymous reviewers. Following their suggestions and comments, we have extensively revised the manuscript and provided a point-to-point response to each comment. The original comments are in **bold** font, our response is in regular font, and the changes in the text are in blue.

**Comment 1**

**Sang et al. studied the interprovincial connections of green water. They quantified this by calculate the precipitation of each province from the green water inside or outside the province. The work is based on the data generated from a previous particle tracking work. Authors connect the results with social-economic effects, which is very interesting and novel. The structure and writing of the paper are clear. However, I have some thoughts as follows:**

**Response:** Thank you for taking your time to review our study and provide feedback and comments. Following the suggestions and comments, we have extensively revised the manuscript and provided a point-to-point response to each comment. The original comments are in **bold** font, our response is in regular font, and the changes in the text are in blue.

1. **I don't think authors have clear enough introduction about how they connect the green water with the social-economic value. The introduction is in lines 140-175, but not clear. There are even no dimensions of the variables, and it is hard to know the relationship between different variables in the equations.**

**Response**:Thank you for the comments.

We apologize for the confusion regarding the introduction about the connections between green water and socio-economic values.

The connection of green water with socio-economic value can be reflected in the cascade that green water from source region forms precipitation in sink region, then precipitation recharges water resources and sustain economic activities, human livelihood and crop growth. We estimated the green water contribution on social-economic values in terms of surface water resource volume, economic output (GDP), population and food production.

In the revision, we added a more detailed introduction about the connections between

green water and socio-economic values in Figure 1 and section Method. Please see the revised texts and figure below.

Green water from upwind source provinces flows and precipitates downwind to recharge water resources, and therefore sustains socio-economic activities in sink provinces, as depicted in Fig. 1. Consequently, precipitation, water resources, and socio-economic factors such as economic activities, human livelihood, and crop production in sink provinces rely on green water exported from source provinces. Changes in green water may affect water resource volume, and then impact economic activities, livelihood, and crop production through water supply.

[Figure]

Figure 1. A conceptual diagram depicts the teleconnection of green water flows and their socio-economic contributions in a cascade manner. Evapotranspiration (green dotted arrows) from source region *i* flows downwind with prevailing winds (green thick arrows) and precipitates in sink region *n*, which recharges water sources and sustains socio-economic activities in sink regions.

2. **It looks like authors assume linear relationships between water and all the social-economic indices. I am not quite sure if this is rigorous. For example, whether the food productivity has the positive, linear relationship with water? Similar question to other social-economic indices.**

**Response:** Thank you for the comments.

We apologize for the confusion regarding the assumption about the relationships between green water and socio-economic indices.

In our assessment, we assume that all socio-economic indices (i.e. water resources volume, economic output, population and food production) are sustained by precipitation originating from green water from source regions. The socio-economic value of ET was calculated using the average values of various socio-economic indicators from 2008 to 2017 to match the climatological moisture trajectories from 2008 to 2017. Any temporal changes in water use efficiency per socio-economic variables are not included in this calculation. Therefore, the positive or negative relationship between socio-economic indicators and water use through time does not affect the value assessment of green water.

We discussed this point in the revision, and the revised texts are shown below.

Moreover, the resulting moisture trajectory data only represent the climatologically average moisture trajectories and ET (Li et al., 2023), neglecting the interannual variability in moisture flow trajectory, e.g., those induced by the influence of extreme weather events or ENSO (Zhao and Zhou, 2021). The interannual variations in green water flow may affect DPR and DSR in some provinces. Human adaptation tends to buffer the impacts of interannual variations on the socio-economy through water resource management such as reservoirs, dams, and other infrastructure. Accounting for interannual variations in green water flows and their socio-economic contribution is worth further investigation.

In future studies, the dynamic linkage between green water, water resources and economic development can be assessed annually by using a long-term moisture tracking dataset with a separation of water sources consumed by socio-economy (surface and groundwater).

3. **For the sections of sources and sinks of green water (sections 3.1 and 3.2), it is hard to say they are really novel as it looks like some known results with a new wrapper. You are talking about the evapotranspiration circulation by adding the 'interprovincial' concept.**

**Response:** Thank you for the comments.

Sections 3.1 and 3.2 present the results of interprovincial green water transfer. Although many research analyzed the spatial pattern of moisture recycling in China from amphoteric and hydrological sciences, they identified moisture source and sinks at the grid (Zhang et al., 2023), river basins (Wang et al., 2023), and ecological regions scale

(Xie et al., 2024). There is a clear understanding of the large-scale spatial pattern of moisture circulation, few studies focus on quantify moisture recycling at the administrative district scales, which is important for the water management. Therefore, this study applies moisture recycling techniques to inform green water transfer at among provinces, which is previously less known but important for regional water resources management.

With this purpose, Sections 3.1 and 3.2 can help water resources managers understand the moisture recycling and the network of interprovincial moisture recycling is the basis of the socio-economic contribution of green water in each province. In the revision, we streamlined 3.1 and 3.2, and merged them. The revised texts are shown below:

**3.1 The interprovincial green water flows in China and their directions**

[revised manuscript text omitted]

**4. Authors said the data are high quality data from previous studies. I think a bit more introduction is necessary.**

**Response**:Thank you for the comments.

In the revision, we added a more detailed introduction about the UTrack moisture tracking dataset. The revised texts are shown below:

This study used the moisture trajectory dataset generated by the Lagrangian moisture tracking model "UTrack-atmospheric-moisture" driven by ERA5 reanalysis data. The model is the state-of-the-art moisture tracking model, producing more detailed evaporation footprints due to the high spatial resolution and reduced unnecessary complexity (Tuinenburg and Staal, 2020). The dataset provides monthly mean moisture flows at the global scale with a spatial resolution of 0.5° for 2008-2017, expressed as the fractions of evaporation from a source grid allocated to precipitation at a sink grid (Tuinenburg et al., 2020). It has been widely used in moisture recycling research with various spatial scales, such as precipitation source of the grid (Staal et al., 2023; Wei et al., 2024; Zhang et al., 2023) and basin scale (Wang et al., 2023), and moisture transport between nations (Rockström et al., 2023). The moisture trajectory dataset was used in

conjunction with the multi-year monthly mean ET of 2008–2017 from the ERA5 reanalysis dataset to estimate precipitation in a sink grid originating from a source grid.

5.  **It is a long-term dataset. So why not analyze the temporal variations of these teleconnections? I think this is more important to audience. The average state is also important, but they are the natural pattern which are caused by the long-term climatic conditions. We should know this basic pattern, but we cannot change it much. The more important thing is the temporal variations which represent the variations caused by some interannual variations of natural conditions or by man-made climate change. This is important to inform the future social-economic development, e.g., if such variations are good or bad and if we need actions to control or facilitate such variations.**

**Response**:Thank you for the comments.

We totally agree with that temporal variations of these teleconnections are very important for the social-economic development and water management. However, the available UTrack moisture tracking dataset only provides multi-year monthly mean state of moisture flow for 2008–2017, preventing us from analyzing interannual variations in moisture trajectory caused by natural or man-made factors. We hope analyze interannual variations of green water teleconnections in subsequent research.

In the revision, we added this point in the section Discussion and the revised texts are shown below.

Moreover, the resulting moisture trajectory data only represent the climatologically average moisture trajectories and ET (Li et al., 2023), neglecting the interannual variability in moisture flow trajectory, e.g., those induced by the influence of extreme weather events or ENSO (Zhao and Zhou, 2021). The interannual variations in green water flow may affect DPR and DSR in some provinces. Human adaptation tends to buffer the impacts of interannual variations on the socio-economy through water resource management such as reservoirs, dams, and other infrastructure. Accounting for interannual variations in green water flows and their socio-economic contribution is worth further investigation.

The tele-connection of green water and socio-economic value is depicted as cascade: green water forms precipitation, then precipitation recharges water resource, which then sustains economic activities, human livelihood and crop growth. So we used the water resource volume, economic output (GDP), population and food production, the four variables, to estimate the green water contribution.

In the revision, we clarify these in section 2.2 and Figure 1. Please see the revised texts and figure below.

Green water from upwind source provinces flows and precipitates downwind to recharge water resources, and therefore sustains socio-economic activities in sink provinces, as depicted in Fig. 1. Consequently, precipitation, water resources, and socio-economic factors such as economic activities, human livelihood, and crop production in sink provinces rely on green water exported from source provinces. Changes in green water may affect water resource volume, and then impact economic activities, livelihood, and crop production through water supply. We chose water resources volume, economic output (measured by GDP), population, and food production as the four socio-economic indictors that are tightly related to water resources to evaluate the socio-economic contributions of green water.

[Figure]

Figure 1. A conceptual diagram depicts the teleconnection of green water flows and their socio-economic contributions in a cascade manner. Evapotranspiration (green dotted arrows) from source region *i* flows downwind with prevailing winds (green thick arrows) and precipitates in sink region *n*, which recharges water sources and sustains socio-economic activities in sink regions.

2.1 We totally agree with that large parts of green water flows probably remain inaccessible for direct human use. Only the water resources generated from precipitation and consumed by human activities sustains the socio-economy. We used the ratio of water resources formed by precipitation (P) in the calculation of green water value. Since this ratio can be eliminated during the calculation process, Equation (1) is

simplified and does not show the ratio involved in the calculation. The process of simplification is below. We take the GDP value of ET in one province as an example:

$$water\ resouces\ value\ of\ green\ water$$
$$= \frac{water\ resources}{\cancel{P}} \times \cancel{P}$$
$$\times\ green\ water\ forming\ P\ fraction$$
$$= water\ resources\ \times green\ water\ forming\ P\ fraction$$

$$GDP\ value\ of\ green\ water$$
$$= \frac{water\ resources\ value\ of\ green\ water}{water\ resources} \times GDP\ value$$
$$= \frac{\cancel{water\ resources}\ \times green\ water\ forming\ P\ fraction}{\cancel{water\ resources}}$$
$$\times\ GDP\ vlalue$$
$$= green\ water\ forming\ P\ fraction\ \times GDP\ value$$

This research assume that all the used water resources are from precipitation generated by green water. This point was added in the Discussion and revised texts are shown below:

Secondly, the socio-economic value assessment of green water in this study only considers green water flows within China, excluding flows moving abroad and to the ocean that may embody socio-economic value beyond the territory of mainland China. We mainly attribute socio-economic values to green water and generated precipitation because precipitation is the ultimate water source for recharging surface and groundwater of a region. Strictly speaking, such attribution needs to be more precise because socio-economy also utilizes streamflow from upstream areas, which deserve separate attention.

In addition, not all water resources replenished by green water-induced precipitation are accessible for human activities since part of them is used by the natural ecosystem (Keys et al., 2019). Therefore, it is necessary to distinguish water sources and consumption to account green water values more accurately. Despite the selected socio-economic indicators closely linked to water resources, green water flows' socio-economic contribution can manifest in other aspects such as livestock production and irrigated agriculture. In future studies, the dynamic linkage between green water, water resources and economic development can be assessed annually by using a long-term moisture tracking dataset with a separation of water sources consumed by socio-economy (surface and groundwater).

2.2 Sections 3.1 and 3.2 show the results of interprovincial green water transfer. Many studies have analyzed the pattern of moisture recycling in China from amphoteric and hydrological sciences, they identified moisture source and sinks at the grid (Zhang et al., 2023), basin (Wang et al., 2023) and ecological regional scale (Xie et al., 2024). These studies present a clear understanding of the large-scale spatial pattern of moisture circulation. Few studies focus on the moisture recycling analysis at the administrative district scale, which is important for the water management. Therefore, this study promotes the application of moisture recycling at the water resources management scale, which is also important for research outside the field of atmospheric science.

Sections 3.1 and 3.2 can help water resources managers understand the moisture recycling. The network of interprovincial moisture recycling is the basis of the socio-economic contribution of green water in each province.

In the revision, we streamlined 3.1 and 3.2, and merged them. The revised texts are shown below.

[revised manuscript text omitted]

**Comment 3**

**The present manuscript provides an emblematic example of integrating green water flows at a sub-national level in water management strategies. It expands on recent studies that highlighted the socio-economic value of green water teleconnections. The topic is suitable for publication and of interest to the readership of EGUsphere.**

**Response:** Thank you for taking your time to review our study and provide feedback and comments. Following the suggestions and comments, we have extensively revised the manuscript and provided a point-to-point response to each comment. The original comments are in **bold** font, our response is in regular font, and the changes in the text are in blue.

**I would recommend the publication of this paper after major revisions. In the following, there are some comments that the authors may want to consider when revising their manuscript. These revisions should enhance the manuscript's clarity and depth.**

**Specific comments:**
**Abstract:**
**Lines 15-16: Pay attention to verb consistency for better clarity and flow.**

**Response:** Thank you for the comments.

We modified this part to be more specific, and the revised texts are shown below:

Green water (terrestrial evapotranspiration) flows from source regions, precipitates downwind via moisture recycling, recharges water resources, and sustains the socio-economy in sink regions.

**Lines 19-21: The dataset used for the analysis is not well introduced or explained. Provide a more detailed and concise explanation of the data used.**

**Response:** Thank you for the comments.

In the revision, we revised the sentence to include more information. The revised texts are shown below:

This study used a climatology mean moisture trajectory dataset produced by the Utrack model for 2008-2017 to quantify interprovincial green water flows in China and their

socio-economic contributions.

And in Section 2.1, we added a more detailed introduction about the UTrack moisture tracking dataset. The revised texts are shown below:

This study used the moisture trajectory dataset generated by the Lagrangian moisture tracking model "UTrack-atmospheric-moisture" driven by ERA5 reanalysis data. The model is the state-of-the-art moisture tracking model, producing more detailed evaporation footprints due to the high spatial resolution and reduced unnecessary complexity (Tuinenburg and Staal, 2020). The dataset provides monthly mean moisture flows at the global scale with a spatial resolution of 0.5° for 2008-2017, expressed as the fractions of evaporation from a source grid allocated to precipitation at a sink grid (Tuinenburg et al., 2020). It has been widely used in moisture recycling research with various spatial scales, such as precipitation source of the grid (Staal et al., 2023; Wei et al., 2024; Zhang et al., 2023) and basin scale (Wang et al., 2023), and moisture transport between nations (Rockström et al., 2023). The moisture trajectory dataset was used in conjunction with the multi-year monthly mean ET of 2008–2017 from the ERA5 reanalysis dataset to estimate precipitation in a sink grid originating from a source grid.

**Line 22: Include the specific value of the average self-recycling ratio.**

**Response:** Thank you for the comments.

In the revision, we added the range of self-recycling ratio, and revised texts are shown below:

Despite self-recycling (ranging from 0.6% to 35%), green water mainly forms precipitation in neighboring provinces, with average interprovincial flow directions from west to east and south to north.

**Lines 32-35: This passage is unclear. Consider rephrasing and supporting it with specific results.**

**Response:** Thank you for the comments.

In the revision, we rephrased this passage, and the revised texts are shown below:

Overall, the embodied socio-economic values of green water flow increase from source to sink provinces, suggesting that green water from less developed provinces effectively supports the higher socio-economic status of developed provinces.

**Introduction:**

**Line 45: Consider adding additional references for the average global terrestrial moisture recycling ratio. Rockström (2023) cites Tuinenburg (2020), they are essentially the same reference.**

**Response:** Thank you for the comments.

In the revision, we changed the references, and the revised texts are shown below:

Terrestrial evapotranspiration (i.e., green water) (Falkenmark and Rockström, 2006), which includes evaporation and transpiration from land and vegetation, contributes to over half of the global precipitation on land (van der Ent et al., 2010; Theeuwen et al., 2023; Tuinenburg et al., 2020).

**Lines 60-61: Clarify the period of reference for the change mentioned. Specify when the change occurred and add recent references for support.**

**Response:** Thank you for the comments.

In the revision, the period of reference was clarified, and the revised texts are shown below:

Due to upstream water withdrawal and dams, global total blue water flow into oceans and internal sinks decreased by 3.5% in 2002 compared to 1961–1990 (Döll et al., 2009).

**Lines 99-103: This section is unclear. Rephrase for better clarity.**

**Response:** Thank you for the comments.

In the revision, we rephrased this passage and revised texts are shown below:

Recent studies analyzed the large-spatial pattern of moisture recycling in China at the grid (Zhang et al., 2023), river basin (Wang et al., 2023), and ecological regions scales (Xie et al., 2024), or for specific regions (Pranindita et al., 2022; Zhang et al., 2024). However, green water flows from different regions are interlinked and become sources and sinks of each other. Such green water transfer at a sub-national scale effectively forms an interconnected green water flow network. It highlights the mutual dependency of green water and its socio-economic contributions, especially for large countries like China.

**General Comments:**

**The Introduction could benefit from clearer explanations of certain passages. Include a characterization of China's moisture recycling patterns, atmospheric**

**circulation, and climatic seasonality to frame the phenomenon of moisture flows. For instance, compare the importance of moisture recycling in China to other regions globally.**

**Response:** Thank you for the comments.

In the revision, we added the introduction about China's moisture recycling patterns, atmospheric circulation, and climatic seasonality. The revised texts are shown below:

The general spatial and seasonal patterns of moisture flows in China are determined by regional atmospheric circulation systems, including prevailing westerly winds (from the west toward the east) in most of China between 30° and 60° (Bridges et al., 2023), the East Asian monsoon in eastern China, and India monsoon in southwestern China. In summer, the East Asian and Indian monsoons supply moisture for precipitation in eastern and southwestern China (Tian and Fan, 2013). In winter, the East Asian monsoon drives northwesterly moisture transport across much of China and generates precipitation (Wu and Wang, 2002).

**Discuss the socio-economic background of the Chinese provinces involved. Highlight key socio-economic sectors and societal issues/characteristics of these regions.**

**Response:** Thank you for the comments.

In the revision, we added the socio-economic background of the Chinese provinces. The revised texts are shown below:

Few studies focus on green water flows at the administrative district scale, which is important for water management. Furthermore, the substantial regional disparities in socio-economic development add complexity to understanding the socio-economic contributions of green water among Chinese provinces. The western provinces with a weak economic status and sparse populations are abundant in water resources (Ya-Feng et al., 2020). In contrast, the economically developed and densely populated eastern provinces suffer from water scarcity (Varis and Vakkilainen, 2001). Therefore, quantifying interprovincial green water flows and evaluating the embedded socio-economic values offer new perspectives for optimizing water resource utilization and mitigating the imbalance in regional socio-economic development.

**Explain why analyzing green water flows at an inter-regional scale is significant, both generally and specifically for China.**

**Response:** Thank you for the comments.

Although many research analyzed the spatial pattern of moisture recycling in China from amphoteric and hydrological sciences, they identified moisture source and sinks at the grid (Zhang et al., 2023), river basins (Wang et al., 2023), and ecological regions scale (Xie et al., 2024). There is a clear understanding of the large-scale spatial pattern of moisture circulation, few researches focus on quantify moisture recycling at the administrative district scales, which is important for the water management. Therefore, this study applies moisture recycling techniques to inform green water transfer at among provinces, which is previously less known but important for regional water resources management.

In the revision, we added the explanation, the revised texts are shown below.

Recent studies analyzed the large-spatial pattern of moisture recycling in China at the grid (Zhang et al., 2023), river basin (Wang et al., 2023), and ecological regions scales (Xie et al., 2024), or for specific regions (Pranindita et al., 2022; Zhang et al., 2024). However, green water flows from different regions are interlinked and become sources and sinks of each other. Such green water transfer at a sub-national scale effectively forms an interconnected green water flow network. It highlights the mutual dependency of green water and its socio-economic contributions, especially for large countries like China. Few studies focus on green water flows at the administrative district scale, which is important for water management. Furthermore, the substantial regional disparities in socio-economic development add complexity to understanding the socio-economic contributions of green water among Chinese provinces. The western provinces with a weak economic status and sparse populations are abundant in water resources (Ya-Feng et al., 2020). In contrast, the economically developed and densely populated eastern provinces suffer from water scarcity (Varis and Vakkilainen, 2001). Therefore, quantifying interprovincial green water flows and evaluating the embedded socio-economic values offer new perspectives for optimizing water resource utilization and mitigating the imbalance in regional socio-economic development.

**Data and Methods:**

**General Comment: This section requires substantial improvements.**

**Structure: Separate the Data and Methods into two subsections. Move Figure 1 to the Methods subsection and provide detailed explanations in the caption.**

**Response:** Thank you for the comments.

In the revision, we separated the Data and Methods into two subsections. And added detailed explanations in the caption of Figure 1. The revised texts are shown below.

[revised manuscript text omitted]

**Figure 1: The caption should be more detailed to enhance understanding.**

**Response:** Thank you for the comments.

In the revision, we added detailed explanations in the caption of Figure 1.

The revised figure and its caption are shown below:

[Figure]

Figure 1. A conceptual diagram depicts the teleconnection of green water flows and their socio-economic contributions in a cascade manner. Evapotranspiration (green dotted arrows) from source region i flows downwind with prevailing winds (green thick arrows) and precipitates in sink region n, which recharges water sources and sustains socio-economic activities in sink regions.

**Lines 127-138: Provide a more detailed explanation of the reconstruction of flows from the UTrack dataset. Clarify the processing with zonal statistics, possibly using equations or schemes for better comprehension.**

**Response:** Thank you for the comments.

In the revision, we added detailed explanations of the reconstruction of flows from the UTrack dataset in Section 2.2 and Figure A1. The revised texts and figure are shown below:

We quantified interprovincial moisture flows and their precipitation contribution following the workflow described in Fig. A1. At each sink grid, the ET to precipitation

fractions from the moisture trajectory datasets were multiplied by ERA5 evapotranspiration (ET) to obtain monthly precipitation contribution by moisture from its source grids. Repeating the calculation for all grids within a sink province and summing them up yielded the precipitation in the sink province contributed by each source grid (Fig. A1 Step 1). Next, we employed zonal statistics to sum up precipitation in the sink province contributed by grids of each source province, and the precipitation contribution was converted to relative values, i.e., the fraction of precipitation in sink province $j$ originating from green water of a source province $i$ (denoted as $W_{ij}$) rather than absolute contribution to reduce the uncertainty in the latter (Fig. A1 Step 2). The fractions $W_{ij}$ multiplied by the observed precipitation of the sink province restore the absolute precipitation contribution. This practice ensures that provincial precipitation is fully decomposed into different sources, avoiding the estimation bias of sink precipitation due to unclosed water balance by ET and precipitation data (De Petrillo et al., 2024). Finally, the interprovincial green water flows in China were derived after estimating each province individually.

[Figure]

**Figure A1.** Workflow of estimating green water flow. Step 1: calculate the precipitation in sink

grids originating from ET in source grids. Step 2: calculate the fraction of precipitation in sink provinces originating from source provinces.

**Socio-economic Analysis: Since this is the core of the study, it needs a more in-depth analysis. Explain the significance of green water flows for the variables considered. How do they contribute to the services these variables represent?**

**Response:** Thank you for the comments.

In the revision, we added the explanation of green water flows' significance. The revised texts are shown below:

Consequently, precipitation, water resources, and socio-economic factors such as economic activities, human livelihood, and crop production in sink provinces rely on green water exported from source provinces. Changes in green water may affect water resource volume, and then impact economic activities, livelihood, and crop production through water supply. We chose water resources volume, economic output (measured by GDP), population, and food production as the four socio-economic indictors that are tightly related to water resources to evaluate the socio-economic contributions of green water.

**Equation 1:**

**Consider incorporating the areal extension of the provinces. Using population density instead of population, and expressing surface water resources per unit area, would be more appropriate. Similarly, express food production per area rather than gross food production. Use GDP per capita (GDP/P) instead of gross GDP.**

**Response:** Thank you for the comments.

The land area and population of source and sink provinces differ, when using per-unit measures, the socio-economic value of green water is affected by the denominators of the indicators for both the source and sink provinces, making the results difficult to interpret. Alternatively, per-unit measures reflect the different characteristics between the source and sink regions, such as the green water flowing from economically less developed to developed provinces indicated by per capita GDP. These results have already been discussed and analyzed in the manuscript in Section Results. Therefore, we chose to use total amounts of socio-economic indicators to calculate the green water value. And those results are shown below:

**3.3 Changing socio-economic values of green water flows**

[Figure]

Figure 5. Changes in socio-economic values embodied in green water flow from source to sink provinces for GDP (a), population (b), and food production (c). Thin arrows of different colors represent the socio-economic value increase (in red) or decrease (in blue) from source to sink provinces. Green bars represent the sum socio-economic value in China's 31 provinces.

The substantial socio-economic values embodied in interprovincial green water flows highlight the teleconnection of green water from source provinces and the socio-economy in sink provinces, including economy, population, and food production. Due to different socio-economic statuses, the same amount of consumed water resources, which are recharged by green water, would sustain different socio-economic values between source and sink provinces. Therefore, the socio-economic values embodied in green water flow would change when traveling from source to sink provinces. As shown in Fig. 5, the socio-economic values embodied in green water flow increase from source to sink provinces by 4 trillion CNY for GDP, 6 million for population, and 25 million tons for food production, respectively. The increase in the embodied GDP, population, and food production is observed in 20, 16, and 22 source provinces among a total of 31. This indicates that green water tends to flow from less to more developed provinces, sustaining more economic production, population, and food production per unit of green water. The largest economic output value increases are in Guangxi (+0.83 trillion CNY, 54%). Xinjiang has the most added value in population (+13 million persons, 59%) and food production (+7 million tons, 60%) because their green water flows to more developed provinces (Fig. A5). In contrast, decreased socio-economic values of green water flow are also observed. Shandong, Shaanxi, and Henan have the largest depreciation in green water values for GDP (-0.66 trillion CNY, 48%), population (-13 million persons, 42%), and food production (-12 million tons, 72%) (Fig. A5) because their green water flows to provinces with lower socio-economic values.

The changing socio-economic values of green water flow reflect the regional disparity

in socio-economic statuses between source and sink provinces. The exported green water for more than half of the source provinces in China (> 15) has increased socio-economic values when reaching sink provinces. This shows that green water from less developed provinces effectively supports the higher socio-economic status of developed provinces through the interprovincial flow network. Therefore, these provinces are vitally important green water providers to developed areas. This teleconnection of green water and socio-economy substantiates that changing land use in the source provinces that affect evapotranspiration is likely to influence water resources availability and socio-economic development in the sink provinces (Dias et al., 2015; Weng et al., 2018). Hence, it is imperative to account for "invisible" green water flow and its cascade effect in large-scale water resources management.

**Address the role of irrigation and irrigation infrastructure in food production to avoid overestimating the contribution of green water flows. Differentiating between irrigated and rainfed productivity could be insightful.**

**Response:** Thank you for the comments.

The irrigation increases ET in source region, then provides more moisture for downwind regions. Simultaneously, most of the water for local irrigation comes from upwind regions. Therefore, it is necessary to distinguish water sources more carefully. In the revision, we added this point in Section Discussion. The revised texts are shown below.

However, the food production data from China Statistical Yearbook is the total food production including both irrigated and rainfed production. It's hard to differentiate the irrigated and rainfed productivity due to the data limitation.

Moreover, the interactions between blue and green water increase the complexity to evaluating green water's socio-economic contribution. For example, the blue water extracted by irrigation increases ET in the source region, providing more moisture for downwind regions (Yang et al., 2019). Simultaneously, most of the blue water for local irrigation comes from the green water of upwind regions (McDermid et al., 2023). In addition, not all water resources replenished by green water-induced precipitation are accessible for human activities since part of them is used by the natural ecosystem (Keys et al., 2019). Therefore, it is necessary to distinguish water sources and consumption to account green water values more accurately.

**Include units of measure.**

**Response:** Thank you for the comments.

In the revision, we added units and the revised texts are shown below:

where $S_j$ is the average socio-economic value of 2008-2017 (i.e., water resources volume (km$^3$), GDP (in unit of CNY, 1 CNY = 0.14 USD), population (persons), and food production (ton)) at sink province $j$, $n$ is the number of sink provinces.

**Equations 2 and 3:**

**The focus shifts to the consumption patterns of each province. However, Equation 1 deals with food production, which does not equate to food consumption. Food production in one province might be exported elsewhere. Clarify whether the study focuses on production, consumption, or both, and how these dynamics are analyzed.**

**Response:** Thank you for the comments.

We apologize for the confusion regarding the research's focus.

Our research focus on the food production. We used water productivity in source province to calculate socio-economic values of its exported green water in the counterfactual scenario when it was all consumed in source province without interprovincial transfer. The results were compared with the actual green water's socio-economic values (namely socio-economic values of exported green water when it is consumed in sink provinces). What we focused is the difference of food production between the scenario that green water is all consumed in source provinces and actual scenario.

In the revision, we clarified this point more clearly. The revised texts are shown below.

Due to the different socio-economic development statuses, the same amount of green water may produce different socio-economic values between source and sink provinces. This means green water flow also involves changes in embodied socio-economic value from source to sink provinces. We used water productivity in the source province ($WP_i$) to calculate the socio-economic values of its exported green water in the counterfactual scenario when it was all consumed in the source province without interprovincial transfer ($GV_i'$) (Eq. 2). The results were compared with the actual green water's socio-economic values (Eq. 1) (namely socio-economic values of exported green water when it is consumed in sink provinces) as:

$$GV_i' = \sum_{j=1}^{n} (W_{i,j} \times WU_j \times WP_i), \tag{2}$$

where $WU_j$ is water use in sink province $j$, and $WP_i$ is water productivity in source province $i$. (i.e., economic output, population, and food production per unit water use).

**General comment: Consider revising Equations 1,2 and 3 to enhance the rigor of the analysis.**

**Results:**

**Section 3.1:**

**Figure 2: The figure has great potential but needs improvements.**

**Increase its size for better readability of numbers and histograms.**

**Clarify the label of the right bar in the figure, caption, and text. Consider rephrasing for better understanding.**

**Response:** Thank you for the comments.

In the revision, we improved Figure 2 to increase its readability. The revised figure is shown below:

[Figure]

Figure 2. Interprovincial green water flows in China. The heat map denotes precipitation in sink province generated by green water from a source province (mm). The right bar shows domestic precipitation (km$^3$) formed by green water from each source province. The top bar shows precipitation in each sink province formed by green water from domestic source provinces (km$^3$).

**Lines 195-237: The discussion on PRR, DPR, and DSR contains a lot of information that is difficult to visualize. Consider creating a figure to represent these results to help the discussion of socio-economic implications.**

**Response:** Thank you for the comments.

The information of PRR, DPR, and DSR in each province has shown in Figure A3 (Appendix Figure 3).

[Figure]

**Figure A2.** (a) Domestic precipitation ratio (DPR), (b) domestic source ratio (DSR) and (c) precipitation recycling ratio (PRR) in each province.

**Line 214: Provide a definition of westerly winds for a general audience. Also, it is the first time in the manuscript that atmospheric circulation is considered explicitly (see comment about the Introduction)**

**Response:** Thank you for the comments.

In the revision, we provided the definition of westerly in Section Introduction. The revised texts are shown below:

The general spatial and seasonal patterns of moisture flows in China are determined by regional atmospheric circulation systems, including prevailing westerly winds (from the west toward the east) in most of China between 30° and 60° (Bridges et al., 2023), the East Asian monsoon in eastern China, and India monsoon in southwestern China.

**Section 3.2:**

**Suggest swapping the order with Section 3.1. The geographic visualization of flows in Section 3.2 aids in understanding the results presented in Section 3.1.**

**Response:** Thank you for the comments.

Section 3.1 shows the network of interprovincial green water flows. Section 3.2 shows the spatial pattern of each province's composited green water flows direction based on the green water flowing network. We think it's more appropriate to clarify the interprovincial green water flowing network firstly.

In the revision, we streamlined 3.1 and 3.2, and merged them. The revised texts are shown below.

**3.1 The interprovincial green water flows in China and their directions**

[revised manuscript text omitted]

**Section 3.3:**

**This section is well-written and interesting. However, given its significance to the analysis, consider expanding and providing more in-depth discussion.**

**Response:** Thank you for the comments.

The in-depth discussion of Section 3.3 is in the Section Discussion. To make this part clearer, we moved the in-depth discussion to Section Results in the revision. The revised texts are shown below.

**3.3 Changing socio-economic values of green water flows**

[Figure]

Figure 5. Changes in socio-economic values embodied in green water flow from source to sink provinces for GDP (a), population (b), and food production (c). Thin arrows of different colors represent the socio-economic value increase (in red) or decrease (in blue) from source to sink provinces. Green bars represent the sum socio-economic value in China's 31 provinces.

The substantial socio-economic values embodied in interprovincial green water flows highlight the teleconnection of green water from source provinces and the socio-economy in sink provinces, including economy, population, and food production. Due to different socio-economic statuses, the same amount of consumed water resources, which are recharged by green water, would sustain different socio-economic values between source and sink provinces. Therefore, the socio-economic values embodied in

green water flow would change when traveling from source to sink provinces. As shown in Fig. 5, the socio-economic values embodied in green water flow increase from source to sink provinces by 4 trillion CNY for GDP, 6 million for population, and 25 million tons for food production, respectively. The increase in the embodied GDP, population, and food production is observed in 20, 16, and 22 source provinces among a total of 31. This indicates that green water tends to flow from less to more developed provinces, sustaining more economic production, population, and food production per unit of green water. The largest economic output value increases are in Guangxi (+0.83 trillion CNY, 54%). Xinjiang has the most added value in population (+13 million persons, 59%) and food production (+7 million tons, 60%) because their green water flows to more developed provinces (Fig. A5). In contrast, decreased socio-economic values of green water flow are also observed. Shandong, Shaanxi, and Henan have the largest depreciation in green water values for GDP (-0.66 trillion CNY, 48%), population (-13 million persons, 42%), and food production (-12 million tons, 72%) (Fig. A5) because their green water flows to provinces with lower socio-economic values.

The changing socio-economic values of green water flow reflect the regional disparity in socio-economic statuses between source and sink provinces. The exported green water for more than half of the source provinces in China (> 15) has increased socio-economic values when reaching sink provinces. This shows that green water from less developed provinces effectively supports the higher socio-economic status of developed provinces through the interprovincial flow network. Therefore, these provinces are vitally important green water providers to developed areas. This teleconnection of green water and socio-economy substantiates that changing land use in the source provinces that affect evapotranspiration is likely to influence water resources availability and socio-economic development in the sink provinces (Dias et al., 2015; Weng et al., 2018). Hence, it is imperative to account for "invisible" green water flow and its cascade effect in large-scale water resources management.

**Discussion:**
**Overall, this section is well-structured and written, but improvements are needed: Enlarge Figure 5 for greater clarity.**
**Response:** Thank you for the comments.
In the revision, we enlarged Figure 5, and the revised figure is shown below:

[Figure]

Figure 5. Changes in socio-economic values embodied in green water flow from source to sink provinces for GDP (a), population (b), and food production (c). Thin arrows of different colors represent the socio-economic value increase (in red) or decrease (in blue) from source to sink provinces. Green bars represent the sum socio-economic value in China's 31 provinces.

**Provide a deeper discussion on the uncertainty of tracked precipitation at the provincial level.**

**Response:** The uncertainty of tracked precipitation includes three aspects: (1) different ET and precipitation datasets lead to different precipitation contribution and PRR due to the ET amount and spatial distribution; (2) the non-closure of the moisture balance between ET and precipitation from ERA5 results in inaccurately capturing actual ET and precipitation volumes; (3) the moisture tracking model has some assumption and simplification. The purpose of our work is not to precisely quantify the moisture recycling quantification but to reveal the relationship between the moisture and socio-economy.

The revised texts in section Discussion are shown below:

Due to the complex dynamics of the green water flow and limitations of the moisture tracking model, there are still major uncertainties in data and methods of this study. First, ET and precipitation datasets driving the UTrack model affect the tracked trajectories and magnitude of moisture flow. The resulting moisture trajectory is expressed as the fraction of ET to precipitation, and the exact amount of moisture is restored by the ET and precipitation datasets chosen by users. Different ET and precipitation datasets could lead to different precipitation contributions and PRR (Li et al., 2023). We used the ERA5 dataset to keep consistent with the original UTrack model. It is noted that the non-closure of the moisture balance from ERA5 (De Petrillo et al., 2024) and simplifications and assumptions introduced in the moisture tracking model

also add uncertainty in the moisture tracking (Tuinenburg and Staal, 2020).

**Lines 429-431: The sentence "Our attempt... [..]" is redundant here and would be more appropriate at the end of the discussion.**

**Response:** Thank you for the comments.

In the revision, we moved this sentence to the end of the discussion.

Nevertheless, our assessment serves as a useful first step to demonstrate the importance of the tele-connected green water flow in addition to blue water. Our attempts to quantify the socio-economy embodied in green water flow fill the gap in green water value assessment and provide a methodological reference for green water management.

**General Comments for Discussion and Conclusions:**

**Include a more in-depth description of the limitations of the socio-economic analysis to add value to these sections and the overall study.**

**Response:** Thank you for the comments.

The limitation of the socio-economic analysis includes:

(1) The green water's socio-economic contribution excludes green water flowing abroad, and did not separate water sources in assessing socio-economic contribution and consider the accessibility of precipitation formed by green water to human activities.

(2) The research selected three socio-economic indices which are tightly linked to the water resources. Other indices linked to water resources like livestock production and irrigated agriculture are not considered.

In the revision, we added these limitations in the section Discussion, and the revised texts are shown below:

Secondly, the socio-economic value assessment of green water in this study only considers green water flows within China, excluding flows moving abroad and to the ocean that may embody socio-economic value beyond the territory of mainland China. We mainly attribute socio-economic values to green water and generated precipitation because precipitation is the ultimate water source for recharging surface and groundwater of a region. Strictly speaking, such attribution needs to be more precise because socio-economy also utilizes streamflow from upstream areas, which deserve separate attention.

In addition, not all water resources replenished by green water-induced precipitation are accessible for human activities since part of them is used by the natural ecosystem (Keys et al., 2019). Therefore, it is necessary to distinguish water sources and

consumption to account green water values more accurately. Despite the selected socio-economic indicators closely linked to water resources, green water flows' socio-economic contribution can manifest in other aspects such as livestock production and irrigated agriculture. In future studies, the dynamic linkage between green water, water resources and economic development can be assessed annually by using a long-term moisture tracking dataset with a separation of water sources consumed by socio-economy (surface and groundwater).

**Discuss potential improvements for this type of socio-economic analysis.**

**Response:** Thank you for the comments.

Potential improvements for this type of socio-economic analysis include (1) the moisture recycling dataset improving; (2) the interannual variations analysis of green water's socio-economic contribution; (3) the distinguish of blue and green water source; (4) more comprehensive assessment of the green water's contribution.

In the revision, we added these potential improvements in the section Discussion, and the revised texts are shown below:

Due to the complex dynamics of the green water flow and limitations of the moisture tracking model, there are still major uncertainties in data and methods of this study. First, ET and precipitation datasets driving the UTrack model affect the tracked trajectories and magnitude of moisture flow. The resulting moisture trajectory is expressed as the fraction of ET to precipitation, and the exact amount of moisture is restored by the ET and precipitation datasets chosen by users. Different ET and precipitation datasets could lead to different precipitation contributions and PRR (Li et al., 2023). We used the ERA5 dataset to keep consistent with the original UTrack model. It is noted that the non-closure of the moisture balance from ERA5 (De Petrillo et al., 2024) and simplifications and assumptions introduced in the moisture tracking model also add uncertainty in the moisture tracking (Tuinenburg and Staal, 2020). Moreover, the resulting moisture trajectory data only represent the climatologically average moisture trajectories and ET (Li et al., 2023), neglecting the interannual variability in moisture flow trajectory, e.g., those induced by the influence of extreme weather events or ENSO (Zhao and Zhou, 2021). The interannual variations in green water flow may affect DPR and DSR in some provinces. Human adaptation tends to buffer the impacts of interannual variations on the socio-economy through water resource management such as reservoirs, dams, and other infrastructure. Accounting for interannual variations in green water flows and their socio-economic contribution is worth further

investigation. Secondly, the socio-economic value assessment of green water in this study only considers green water flows within China, excluding flows moving abroad and to the ocean that may embody socio-economic value beyond the territory of mainland China. We mainly attribute socio-economic values to green water and generated precipitation because precipitation is the ultimate water source for recharging surface and groundwater of a region. Strictly speaking, such attribution needs to be more precise because socio-economy also utilizes streamflow from upstream areas, which deserve separate attention.

Moreover, the interactions between blue and green water increase the complexity to evaluating green water's socio-economic contribution. For example, the blue water extracted by irrigation increases ET in the source region, providing more moisture for downwind regions (Yang et al., 2019). Simultaneously, most of the blue water for local irrigation comes from the green water of upwind regions (McDermid et al., 2023). In addition, not all water resources replenished by green water-induced precipitation are accessible for human activities since part of them is used by the natural ecosystem (Keys et al., 2019). Therefore, it is necessary to distinguish water sources and consumption to account green water values more accurately. Despite the selected socio-economic indicators closely linked to water resources, green water flows' socio-economic contribution can manifest in other aspects such as livestock production and irrigated agriculture. In future studies, the dynamic linkage between green water, water resources and economic development can be assessed annually by using a long-term moisture tracking dataset with a separation of water sources consumed by socio-economy (surface and groundwater).

**Since this study is presented as a starting example of integrating green water teleconnections into water management strategies for socio-economic applications, it would be beneficial to elaborate on additional steps needed to achieve this goal.**
**Response:** Thank you for the comments.

In the revision, we added these potential improvements in the section Discussion, and the revised texts are shown below:

In future studies, the dynamic linkage between green water, water resources and economic development can be assessed annually by using a long-term moisture tracking dataset with a separation of water sources consumed by socio-economy (surface and groundwater). Nevertheless, our assessment serves as a useful first step to demonstrate the importance of the tele-connected green water flow in addition to blue water. Our

attempts to quantify the socio-economy embodied in green water flow fill the gap in green water value assessment and provide a methodological reference for green water management.

**Consider discussing other variables that could enhance the analysis of socio-economic implications.**

**Response:** Thank you for the comments.

Some other socio-economic variables like livestock and irrigated agriculture may enhance the analysis. In the revision, we added them and the revised texts are shown below:

Despite the selected socio-economic indicators closely linked to water resources, green water flows' socio-economic contribution can manifest in other aspects such as livestock production and irrigated agriculture.

**Supplementary Figures and Tables are not cited, and thus integrated in the text. Please integrate them in the main text.**

**Response:** Thank you for the comments.

We apologize for the confusion regarding the figures citing and Supplementary Figures and Tables citing. In the main text, supplementary figures were cited as Fig. A1, Fig. A2; supplementary tables were cited as Table A1 and Table A2. All supplementary figures and tables were cited in section Result and Discussion.

**Reference:**

[revised manuscript text omitted]

---

## Referee Report (RR1)

Firstly, I would like to acknowledge all the work that you put in to revise the manuscript. I think that your revision of the manuscript addresses most of the previously raised comments adequately and clarifies the overall methods and outcome of the study.

(1) However, there is still a misunderstanding in the closure of the water balance in the use of the UTrack dataset, which becomes apparent in lines 155-171 of the revised manuscript. By converting the tracked volumes back into ratios and then using provincial P you are indeed making sure that there is the data-given amount of P. However, this quick-fix only makes sure that there is not too much or too little water but it does not correct the allocation of water from ET from the source cells. Therefore, stating that this fully avoids the bias in the estimation (lines 167-170) seems to me as an overstatement that can lead to misunderstandings. I urge you to correct this by changing the phrasing of the sentence (e.g. in line 168 from 'avoiding the estimation bias' to 'decreasing the estimation bias'). While this may seem like an almost indifferentiable change, it ensures that it is clearer that the bias in the trajectories is still there. Moisture tracking model, in general, still diverge significantly in their estimations, even with the exact same forcing, making it even more important that the bias in a 'single-model estimation' needs to be treated right and communicated clearly.

Very minor comment:

(2) In line 156 you mention the abbreviation 'ET' before its explanation in line 158. Also, you could consider using P as abbreviation for precipitation to stay consistent.

---

## Author Response (AR2)

We deeply appreciate the detailed and constructive comments provided by anonymous reviewers. Following their suggestions and comments, we have extensively revised the manuscript and provided a point-to-point response to each comment. The original comments are in **bold** font, our response is in regular font, and the changes in the text are in blue.

**Report 1**

**Firstly, I would like to acknowledge all the work that you put in to revise the manuscript. I think that your revision of the manuscript addresses most of the previously raised comments adequately and clarifies the overall methods and outcome of the study.**

**(1) However, there is still a misunderstanding in the closure of the water balance in the use of the UTrack dataset, which becomes apparent in lines 155-171 of the revised manuscript. By converting the tracked volumes back into ratios and then using provincial P you are indeed making sure that there is the data-given amount of P. However, this quick-fix only makes sure that there is not too much or too little water but it does not correct the allocation of water from ET from the source cells. Therefore, stating that this fully avoids the bias in the estimation (lines 167-170) seems to me as an overstatement that can lead to misunderstandings. I urge you to correct this by changing the phrasing of the sentence (e.g. in line 168 from 'avoiding the estimation bias' to 'decreasing the estimation bias'). While this may seem like an almost in differentiable change, it ensures that it is clearer that the bias in the trajectories is still there. Moisture tracking model, in general, still diverge significantly in their estimations, even with the exact same forcing, making it even more important that the bias in a 'single-model estimation' needs to be treated right and communicated clearly.**

**Response**:Thank you for the comments.

We acknowledge that the non-closure of water balance in the UTrack dataset due to the non-closure between precipitation and evaporation data from ERA5 dataset is not addressed in this manuscript. Using the provincial precipitation data can't avoid the bias in allocation of ET from source cells. The statement in our previous manuscript was inappropriate. The revised texts are shown below:

This practice ensures that provincial precipitation is fully decomposed into different sources, reducing the estimation bias of sink precipitation due to unclosed water balance

by ET and precipitation data (De Petrillo et al., 2024).

In this study, the non-closure of water balance in the UTrack dataset impacts the green water flow trajectory, and affects the share of precipitation contribution from each province's green water to others. This point was added in the Discussion. The revised texts are shown below:

It is noted that the non-closure of the hydrological balance from ERA5 (De Petrillo et al., 2024) and divergence in moisture tracking models (e.g., simplifications and assumptions) also add uncertainty and impact the accuracy of the tracked green water flow (Tuinenburg and Staal, 2020; Zhang et al., 2023).

**Very minor comment:**

**(2) In line 156 you mention the abbreviation 'ET' before its explanation in line 158. Also, you could consider using P as abbreviation for precipitation to stay consistent.**

**Response:** Thank you for the comments.

We added the explanation of the abbreviation 'ET' in line 156, and removed the explanation in line 158. The revised texts are shown below:

At each sink grid, the evapotranspiration (ET) to precipitation (ET-to-P) fractions from the moisture trajectory datasets were multiplied by ERA5 ET to obtain monthly precipitation contribution by moisture from its source grids.

We used the abbreviation of precipitation in the special variable "ET-to-P". We think it's unnecessary to use the abbreviation elsewhere.

**References:**

De Petrillo, E., Fahrländer, S., Tuninetti, M., Andersen, L. S., Monaco, L., Ridolfi, L., and Laio, F.: Reconciling tracked atmospheric water flows to close the global freshwater cycle, https://doi.org/10.21203/rs.3.rs-4177311/v2, 30 April 2024.

Tuinenburg, O. A. and Staal, A.: Tracking the global flows of atmospheric moisture and associated uncertainties, Hydrology and Earth System Sciences, 24, 2419–2435, https://doi.org/10.5194/hess-24-2419-2020, 2020.

Zhang, C., Chen, D., Tang, Q., and Huang, J.: Fate and Changes in Moisture Evaporated From the Tibetan Plateau (2000–2020), Water Resources Research, 59, e2022WR034165, https://doi.org/10.1029/2022WR034165, 2023.